# Room-temperature quantum interference in single perovskite quantum dot junctions

Haining Zheng[1,4], Songjun Hou[2,4], Chenguang Xin[3,4], Qingqing Wu[2], Feng Jiang[1], Zhibing Tan[1], Xin Zhou[3], Luchun Lin 📷 [1], Wenxiang He[1], Qingmin Li[1], Jueting Zheng[1], Longyi Zhang[1], Junyang Liu[1], Yang Yang[1], Jia Shi[1], Xiaodan Zhang[3], Ying Zhao[3], Yuelong Li[3]*, Colin Lambert 📷 [2]* & Wenjing Hong 📷 [1]*

The studies of quantum interference effects through bulk perovskite materials at the Ångstrom scale still remain as a major challenge. Herein, we provide the observation of room-temperature quantum interference effects in metal halide perovskite quantum dots (QDs) using the mechanically controllable break junction technique. Single-QD conductance measurements reveal that there are multiple conductance peaks for the $CH_3NH_3PbBr_3$ and $CH_3NH_3PbBr_{2.15}Cl_{0.85}$ QDs, whose displacement distributions match the lattice constant of QDs, suggesting that the gold electrodes slide through different lattice sites of the QD via Au-halogen coupling. We also observe a distinct conductance 'jump' at the end of the sliding process, which is further evidence that quantum interference effects dominate charge transport in these single-QD junctions. This conductance 'jump' is also confirmed by our theoretical calculations utilizing density functional theory combined with quantum transport theory. Our measurements and theory create a pathway to exploit quantum interference effects in quantum-controlled perovskite materials.

[1] State Key Laboratory of Physical Chemistry of Solid Surfaces, iChEM, NEL, College of Chemistry and Chemical Engineering, Xiamen University, Xiamen 361005, China. [2] Department of Physics, Lancaster University, Lancaster LA1 4YB, UK. [3] Institute of Photoelectronic Thin Film Devices and Technology, Key Laboratory of Photoelectronic Thin Film Devices and Technology of Tianjin, Key Laboratory of Optical Information Science and Technology of Ministry of Education, Nankai University, Tianjin 300350, China. [4] These authors contributed equally: Haining Zheng, Songjun Hou, Chenguang Xin. *email: lyl@nankai.edu.cn; c.lambert@lancaster.ac.uk; whong@xmu.edu.cn

Quantum interference (QI) effects, originating from de Broglie waves of electrons traversing different pathways through nanoscale junctions, underpin the conceptual designs of molecular devices such as QI based field-effect transistors[1,2]. Previous experimental and theoretical investigations of room-temperature QI effects have mainly focused on organic molecular wires, including π-conducting wires[3], σ-conducting wires[4], and even π-stacked dimers[5], but the exploitation of room-temperature QI effects in electron transport through Ångstrom-scale inorganic systems still remains unexplored. The unique quantum yields and high carrier mobility of perovskite-based electronic materials offer a platform for us to translate knowledge of their macroscopic charge transport into the quantum effects at the nanoscale.

Perovskite materials attract extraordinary attention in applications of the light-emitting diode[6], photodetector[7], and solar cells[8,9]. Although there are many experimental investigations of charge transport through bulk perovskite materials, including thin films[10], nanocrystals[11], and single crystals[12], investigations at the nanoscale, to reveal QI effects in their room-temperature transport properties remain as a major experimental challenge. The extensions of single-molecule charge transport measurements from conjugated molecular families[13] to molecular assemblies[14,15], clusters[16], and the recently developed Au-halogen interfacial engineering[17] offer an opportunity to gain an insight into microscopic charge transport through Ångstrom-scale perovskite materials.

To understand how their macroscopic charge transport properties lead to quantum effects at the nanoscale, here we report the observation of room-temperature QI effects in metal halide perovskite quantum dots (QDs) at the Ångstrom scale using the mechanically controllable break junction (MCBJ) technique combined with quantum transport theory and calculations. Multiple distinguishable conductance peaks can be observed in the MAPbBr$_3$ and MAPbBr$_{2.15}$Cl$_{0.85}$ (MA = CH$_3$NH$_3^+$) QDs, while MAPbBr$_{2.15}$I$_{0.85}$, MAPbCl$_3$ and MAPbI$_3$ QDs show no significant conductance features. The displacement distributions also match well with the lattice constant of QDs, suggesting that the multiple conductance features are derived from the sliding of gold electrodes through different lattice sites of the QD via Au-halogen coupling. A distinct conductance 'jump' is also observed at the end of the sliding process, which is further evidence that QI effects dominate charge transport in the single-QD junctions.

## Results

**Theory of room-temperature QI effects.** The tunneling transport through QDs or molecules is mediated by electrons whose energy lies within the energy gap between the highest occupied molecular orbital (HOMO) and lowest occupied molecular orbital (LUMO), thus the inter-orbital QI can be understood qualitatively by inspecting the signs of the HOMO and LUMO at the points of contact between the molecule and electrode (Fig. 1a). As mentioned in previous literature[18–20], if the coupling between molecule and electrode is weak, the effect of QI on transport properties could be predicted by examining the Green's function $G(E_F)$ of the isolated molecule. The transmission amplitude of an electron with energy $E_F$ from site $i$ to $j$ is proportional to $G_{i,j}(E_F) = \sum_{n=1}^{N} \frac{\phi_i^n \phi_j^n}{E_F - \varepsilon_n}$, where $\phi_i^n$ is the amplitude of $n$th molecular orbital (MO) on site $i$ and $\varepsilon_n$ is the corresponding MO energy level. Taking only the HOMO and LUMO into consideration and assuming that $E_F$ is located in the midgap of HOMO and LUMO, this equation could be further written as $G_{i,j}(E_F) \approx \frac{1}{\Delta}\left(\phi_i^{HOMO}\phi_j^{HOMO} - \phi_i^{LUMO}\phi_j^{LUMO}\right) = \frac{1}{\Delta}(a_H - a_L)$, where $\Delta$ is half of the gap of HOMO and LUMO, $a_H = \phi_i^{HOMO}\phi_j^{HOMO}$, $a_L = \phi_i^{LUMO}\phi_j^{LUMO}$. Therefore, constructive quantum interference

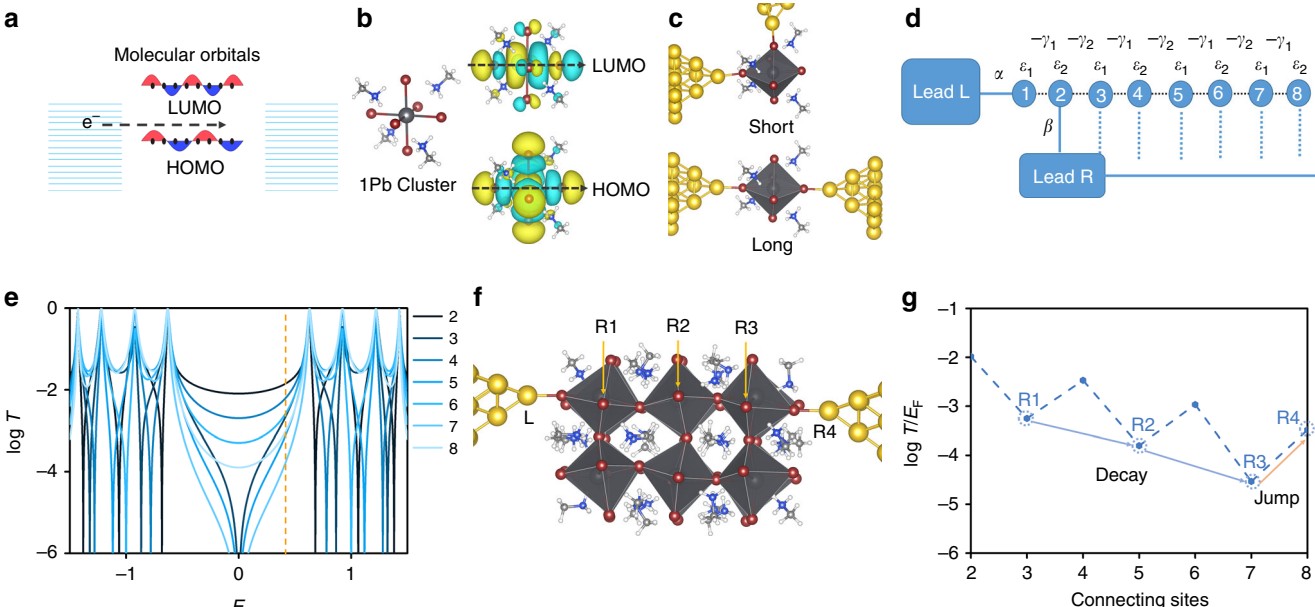

**Fig. 1** The orthogonality of molecular orbitals and QI of 1Pb perovskite cluster. **a** Schematic of coherent tunneling across a molecule, where the HOMO and LUMO of a one-dimensional chain of 8 sites are plotted. **b** The chemical structure of a relaxed 1Pb MAPbBr$_3$ (MA = CH$_3$NH$_3^+$) cluster, and its corresponding HOMO and LUMO. The number of nodes is 4 and 5 in the direction indicated by the black dashed lines. **c** A relaxed 1Pb MAPbBr$_3$ cluster is embedded between two gold electrodes, with two different connections, denoted 'Short' and 'Long' separately. Pb, Br, N, C, H, and Au atoms are represented by large gray, purple, blue, small gray, white, and yellow balls. **d** Schematic of tight-binding model comprising an 8-site diatomic chain as the scattering region, where site 1 is connected to left lead L and sites 2 to 8 are connected sequentially to right lead R. **e** The corresponding transmission functions when lead R is attached different sites from 2 to 8. The Fermi energy is indicated by the yellow line. **f** Relaxed conformation for a 12Pb MAPbBr$_3$ cluster attached to two gold electrodes. The Br atom connected to left lead is labeled as 'L', while the Br atoms attached to right lead are labeled 'R1', 'R2', 'R3', and 'R4'. **g** Transmission function T(EF) at the Fermi energy for different sites connected to the right lead.

(CQI) corresponding to a large value of $|G_{i,j}(E_F)|$ is predicted if $a_H$ and $a_L$ have opposite signs, while destructive quantum interference (DQI), corresponding to a low value of $|G_{i,j}(E_F)|$, is predicted if $a_H$ and $a_L$ have the same sign. As an example of this sign dependence, if the electrodes make contact with the left and right ends of the perovskite cluster in Fig. 1c, the LUMO has a positive amplitude on the left (yellow) and a negative amplitude on the right (blue), hence the product ($a_L$) is negative. On the other hand, the HOMO has a positive amplitude on the left and a positive amplitude on the right, hence the product ($a_H$) is positive. Therefore, CQI is expected. Hence when electrodes are attached to the perovskite junctions with 'long' or 'short' sites in Fig. 1c, the transmission functions obtained from density functional theory (DFT) calculation reveal counter-intuitively that the conductance of the latter is higher than that of the former over a wide energy range (see Supplementary Fig. 24).

The fact that HOMO and LOMO orbital products corresponding to contacts at the ends of such molecules are of opposite signs is a consequence of orthogonality. The reason is that orthogonality requires that the number of sign changes must differ by unity if the nodal structure of the HOMO and LOMO are the same in the direction transverse to their long axis. Therefore CQI is expected to be a common feature of end-contacted molecules. As shown in Fig. 1d, if one electrode is placed at the left end of a molecule (site 1) and the other electrode makes successive contacts along the length of a molecule ($L = 2, 3, \ldots\ldots 8$) (the other models are shown in Supplementary Fig. 23), counter-intuitively, the conductance measured at the largest value of $L$ (site 8) should lie above the trend defined by the tunneling decay equation $G \sim e^{-\beta L}$. The transmission function for this simple

model (Fig. 1e) shows that quantum oscillations will occur over a wide range of electron energies $E$ within the HOMO-LUMO gap under these circumstances. To employ this model for the perovskite materials, as indicated in Fig. 1f, a model for contacting perovskite clusters involves successive contacts with odd-numbered sites, followed by a conductance jump at the final contact (orange arrow in Fig. 1g). The above analysis suggests that perovskite quantum clusters provide an ideal platform for identifying room-temperature QI transport features at the Ångstrom scale.

**Single-molecule conductance measurements.** To explore the QI in perovskite clusters, we experimentally investigate electron transport through single perovskite QD junctions bonded to two gold electrodes through Au-halogen bonds. Four types of organic-inorganic metal halide perovskite QDs MAPbX$_3$ (MA = CH$_3$NH$_3^+$, $X = I^-$, Br$^-$, Cl$^-$, a mixture of Br$^-$ and Cl$^-$) are synthesized with oleic acid and octylamine as ligands to enhance colloidal stability and suppress QD aggregation effects (See Method, Supplementary Figs. 5–6 and Supplementary Note 2–3 for more details)[21]. As shown in Fig. 2a, the typical ABX$_3$ perovskite-type structure is composed of the framework of [PbX$_6$]$^-$ octahedra occupied by methylammonium cation (MA$^+$) in the four octahedra central positions. Single-QD conductance measurements of MAPbBr$_3$ are carried out using the MCBJ technique in a solution containing 0.365 mg mL$^{-1}$ QDs with 1, 3, 5-trimethylbenzene (TMB) as a solvent (see Supplementary Figs. 7–8 and Note 4 for more details of the MCBJ measurement)[22]. As shown in Fig. 2b, the individual conductance-distance curves of solvent without QDs show a monotonic

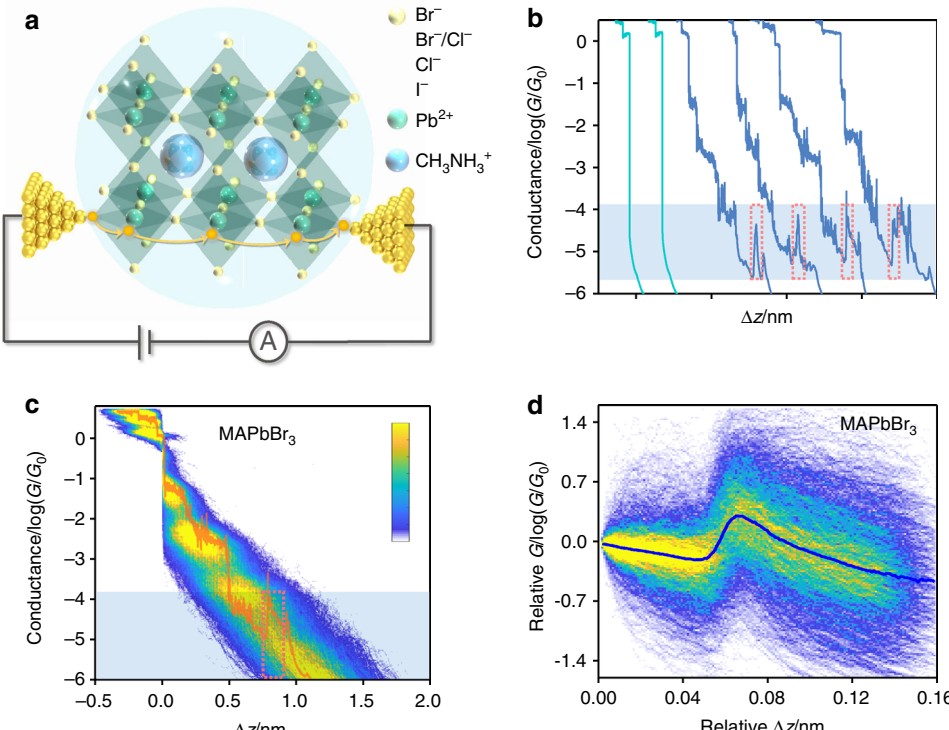

**Fig. 2** MCBJ measurements of MAPbBr$_3$ QDs. **a** Schematic of the MCBJ experimental principle in MAPbX$_3$ QDs. (MA = CH$_3$NH$_3^+$, $X = I^-$, Br$^-$, Cl$^-$, mixture of Br$^-$ and Cl$^-$.) The yellow arrows indicate the sliding process of the gold electrodes during the MCBJ measurements. **b**, Typical individual conductance-distance traces of pure solvent (green) and MAPbBr$_3$ QDs (blue) (the other conductance-distance traces are shown in Supplementary Fig. 13). The jump plateaus are shown by red frame. **c** All-data-points 2D conductance versus relative distance ($\Delta z$) histogram of MAPbBr$_3$ QDs (approximately 3400 traces) and selected one conductance-distance trace. **d** 2D relative conductance ($G$) verse relative displacement ($\Delta z$) histogram of the 'jump curves' (approximately 2400 traces) at the conductance-distance region marked in Fig. 2c.

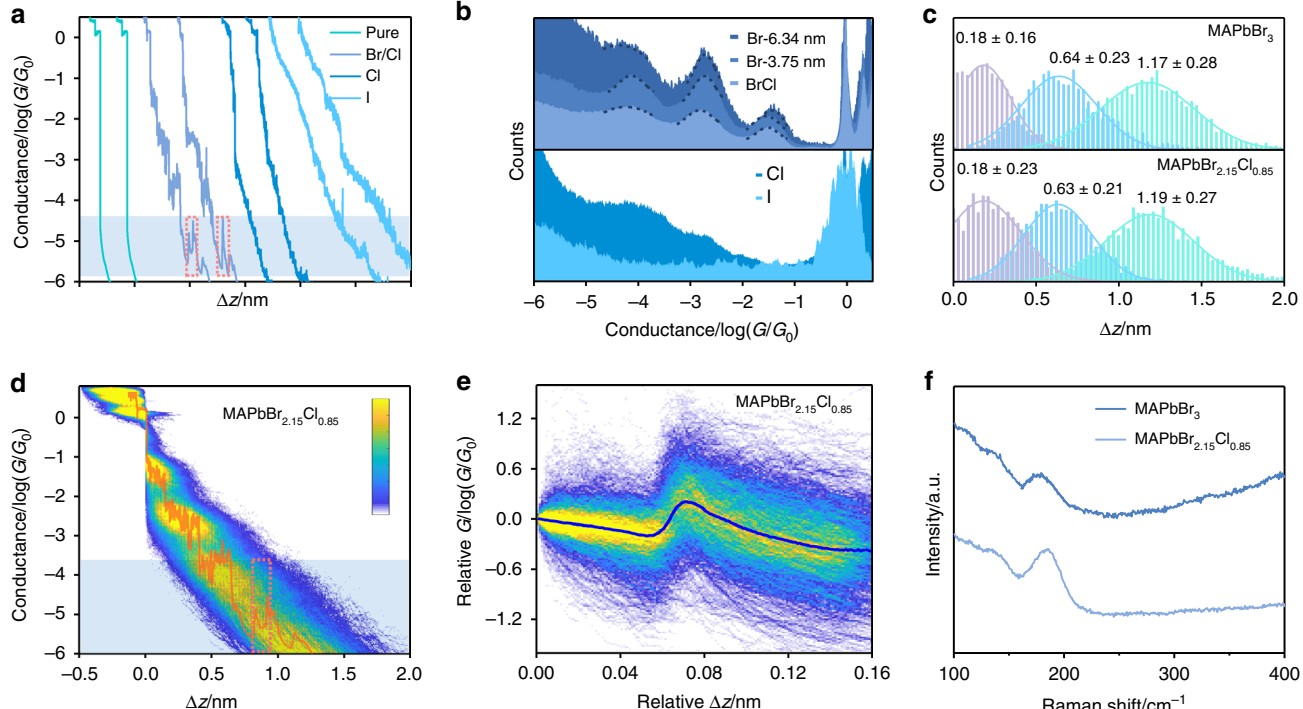

**Fig. 3** MCBJ measurements of MAPbBr$_3$, MAPbBr$_{2.15}$Cl$_{0.85}$, MAPbCl$_3$, and MAPbI$_3$ QDs. **a** Typical individual conductance-distance traces of pure solvent, MAPbBr$_{2.15}$Cl$_{0.85}$, MAPbCl$_3$, and MAPbI$_3$. **b** 1D Conductance histogram constructs without data selection for MAPbBr$_3$, MAPbBr$_{2.15}$Cl$_{0.85}$, MAPbCl$_3$ and MAPbI$_3$ QDs. The average diameters of MAPbBr$_3$ QDs are 6.34 nm (Supplementary Fig. 3b) and 3.75 nm (Supplementary Fig. 2), respectively. The conductance-distance traces are recorded approximately 2500 traces. **c** The displacement distributions of three plateaus for MAPbBr$_3$ (up) and MAPbBr$_{2.15}$Cl$_{0.85}$ (bottom). **d** All-data-points 2D conductance versus relative distance ($\Delta z$) histogram for MAPbBr$_{2.15}$Cl$_{0.85}$ and selected individual conductance-distance trace. **e** 2D relative conductance ($G$) verse relative displacement ($\Delta z$) histogram of the 'jump curves' (approximately 1400 traces) for MAPbBr$_{2.15}$Cl$_{0.85}$. **f** Raman spectra of Au–Br interaction on the gold substrates with SHINERS nanoparticles.

exponential decay after the breaking of gold-gold atomic junctions at conductance quantum $G_0$ (where $G_0$ is the conductance quantum, which equals $2e^2$ h$^{-1}$), while three distinguished conductance plateaus and 'jump plateaus' appear in the traces of MAPbBr$_3$. To reveal the source of the conductance plateaus, we also carry out the MCBJ measurements of all ligands and ingredients used in the synthesis of the QDs in the solvent γ-butyrolactone, including oleic acid, octylamine, PbBr$_2$, PbCl$_2$, MACl, and MABr. The obvious conductance plateau can be observed in PbBr$_2$, while no clear conductance signal can be observed in other ligands and raw materials, suggesting that the conductance signal may come from Au–Br interaction and the other ligands cannot form the single-QD junction (Supplementary Fig. 10 and Note 6). We also characterize the bias-voltage dependence of single-QD conductance over the range from 50 to 250 mV (See Supplementary Fig. 11a), which agrees with the Simmons model and suggests that charge transport is mediated by an off-resonant coherent tunneling mechanism. The single-QD junctions become quite unstable at higher bias voltage (300 and 400 mV) and the conductance values of three plateaus are difficult to identify, which may be due to the destruction of the perovskite clusters at such high electric fields (See Supplementary Fig. 12 and Note 8).

To further demonstrate the conductance evolution during the break junction processes, the two-dimensional (2D) conductance-displacement histogram is plotted in Fig. 2c, and shows multiple distinct conductance clouds, indicating a high molecular junction formation probability and distinct charge transport properties of each configuration. Interestingly, we observe a clear conductance jump at the end of the third plateau in approximately 70% of the

individual conductance-distance traces of MAPbBr$_3$ (See Supplementary Fig. 14 and Note 9 for more analytical details)[23]. As shown in Fig. 2d, a clear jump in conductance could be observed at the relative displacement of approximately 0.05 nm with conductance difference around one order of magnitude, suggesting that the single-QD junction exhibits higher conductive state at the end of the sliding process of the two gold electrodes on the QDs.

To reveal the binding geometries of the single-QD junctions, we carry out the single-QD conductance measurements of MAPbBr$_{2.15}$Cl$_{0.85}$, MAPbBr$_{2.15}$I$_{0.85}$, MAPbCl$_3$ and MAPbI$_3$ QDs. Figure 3a shows several individual conductance-distance curves of these three QDs and pure solvent. For MAPbBr$_{2.15}$Cl$_{0.85}$, multiple conductance features are also observed, which are similar to those of MAPbBr$_3$. The one-dimensional (1D) conductance histograms (Fig. 3b) also show three similar conductance features located at $10^{-1.54}$, $10^{-2.72}$ and $10^{-4.13}G_0$ for MAPbBr$_3$, $10^{-1.51}$, $10^{-2.81}$ and $10^{-4.21}G_0$ for MAPbBr$_{2.15}$Cl$_{0.85}$, respectively, and the conductance evolution and 'jump curves' is also similar with that of MAPbBr$_3$ (see Fig. 3d, e, Supplementary Fig. 15 and Note 10), suggesting the binding of MAPbBr$_{2.15}$Cl$_{0.85}$ also comes from the Au–Br coordination. We also construct the conductance histograms for MAPbCl$_3$, MAPbI$_3$, and MAPbBr$_{2.15}$I$_{0.85}$ from approximately 2500 individual traces (Fig. 3b and Supplementary Fig. 16), and no conductance peaks are observed, while the peak of the gold–gold atomic junction at $G_0$ for MAPbI$_3$ and MAPbBr$_{2.15}$I$_{0.85}$ becomes less clear than others. In addition, we also carry out the MCBJ measurements using the MAPbBr$_3$ QDs with the average diameters of 6.34 nm and 3.75 nm, which are obtained from the

centrifugal speeds of 10000 rpm and 5000 rpm, respectively (as shown in Supplementary Figs. 3 and 17). The experimental results show that the QDs with different diameters show similar conductance features, indicating that the conductance plateaus we measured originate from the perovskite crystal cells rather than the entire perovskite QDs. Furthermore, we calculate the Au-halogen binding energy by using DFT and find that the Au-halogen binding energy is in accordance with the order of Au-I > Au–Br > Au–Cl (See Supplementary Table 3 and Note 16 for more details). The comparison of different QDs suggests that for MAPbCl$_3$ QDs, the bond energy of Au–Cl bond is too weak to form stable Au-QD-Au junctions. In contrast, the strong Au–I bond may break the crystal structure of MAPbBr$_{2.15}$I$_{0.85}$ and MAPbI$_3$ with the sliding process of the electrodes due to the poorer stability of crystal structure[24–26].

To understand the origins of the multiple conductance features, we analyze the relative displacement distribution of MAPbBr$_3$ and MAPbBr$_{2.15}$Cl$_{0.85}$ QDs (The detailed analysis of how to obtain the displacement distribution is discussed in Supplementary Fig. 9 and Note 5). As shown in Fig. 3c, for MAPbBr$_3$ QDs, the most probable displacements of each conductance features are 0.18 ± 0.16 nm, 0.64 ± 0.23 nm and 1.17 ± 0.28 nm, while the displacements are 0.18 ± 0.23 nm, 0.63 ± 0.21 nm and 1.19 ± 0.27 nm for MAPbBr$_{2.15}$Cl$_{0.85}$ QDs, respectively. The average displacement differences are determined to be approximately 0.5 nm from the difference of the above values, which are quite similar for both QDs. The difference of adjacent statistical lengths is approximately consistent with the adjacent lattice distance of Br, confirming that it is the Au–Br coordination that provides the binding sites for Au-QD-Au junctions, during the sliding of gold electrode across the QD's surface. Charge transport investigation of halogen-terminated single-molecule

oligothiophene junctions also suggested that Au-halogen inter-action could act as a robust anchoring group for binding the molecules to the gold electrodes[17]. As for the other atoms, the MA$^+$ is located at the center of the regular octahedron, which is not easy to connect to the gold electrodes, and the adjacent distance of the MA$^+$ is not in accordance with the displacement distributions. The electronegativity of the Pb$^{2+}$ is low, and the Pb$^{2+}$ is hidden within the Br networks that could not have reliable interaction with the gold electrodes. Therefore, the gold electrodes interact with halogen, rather than other atoms or groups, to form stable Au-QD-Au junctions. To provide direct evidence of the Au–Br bond, we further perform the shell-isolated nanoparticle-enhanced Raman spectroscopy (SHINERS). As shown in Fig. 3f, two distinct Raman peaks can be observed at approximately 180 cm$^{-1}$ for MAPbBr$_3$ and MAPbBr$_{2.15}$Cl$_{0.85}$, which confirms the formation of Au–Br bond[27,28], (experimental details are shown in Supplementary Note 11) suggesting the multiple conductance features originate from the sliding of gold electrodes on the surface of single MAPbBr$_3$ and MAPbBr$_{2.15}$Cl$_{0.85}$ QD via the Au–Br bond.

**DFT calculation.** To gain further insight into the conductance trends observed in the MCBJ measurements, transmission spectra $T(E)$ are calculated by combining the DFT package SIESTA[29] with the quantum transport code Gollum[30] (see Method for further details). The MAPbBr$_3$ neutral charge clusters (1Pb (MA$_4$PbBr$_6$), 8Pb (MA$_{20}$Pb$_8$Br$_{36}$), 12Pb (MA$_{28}$Pb$_{12}$Br$_{52}$), 16Pb (MA$_{36}$Pb$_{16}$Br$_{68}$)) are built with the same method as the literature[31] (see Supplementary Fig. 25). For the crystal MAPbBr$_3$, our calculated band gap of 2.31 eV agrees well with the experimental value 2.24 eV (Fig. 4a)[32], along with the

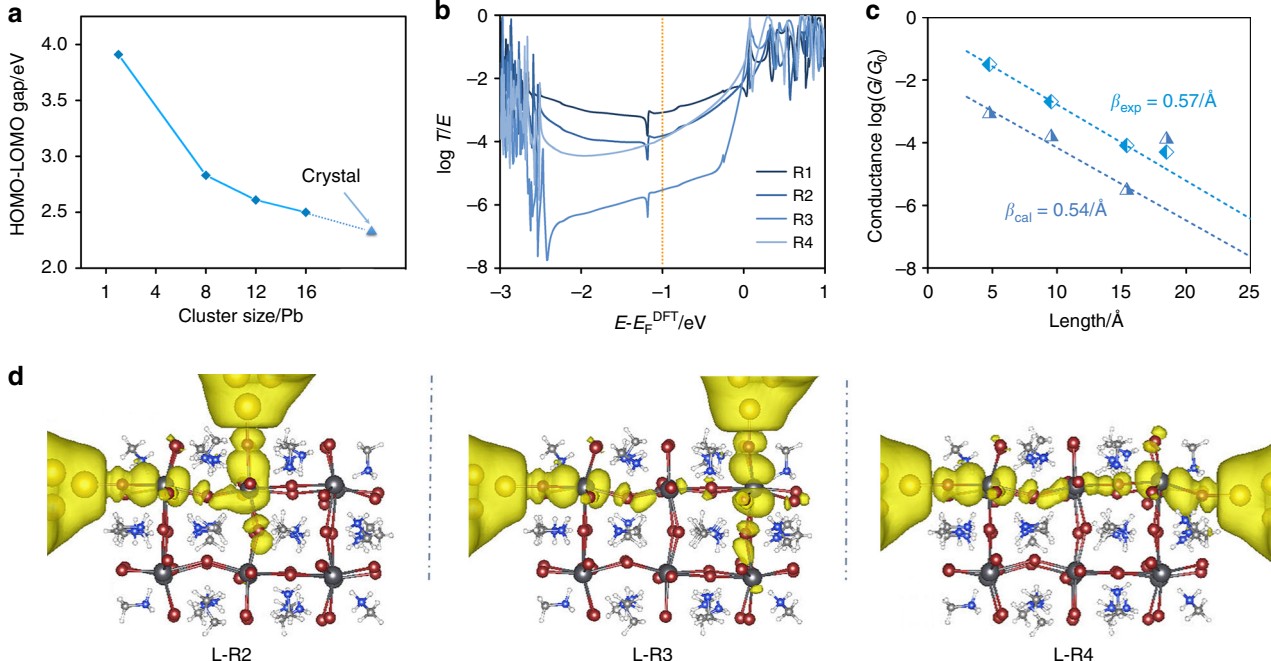

**Fig. 4** The charge transport property of 12Pb MAPbBr$_3$ with different connectivities. **a** HOMO-LUMO gaps with respect to the size of MAPbBr$_3$ clusters. **b** The transmission spectra of different connectivities as the function of $E - E_{\mathrm{F}}^{\mathrm{DFT}}$. **c** The corresponding experimental and theoretical conductance evolution versus the increasing separation between the two electrodes. The theoretical dots stand for the room-temperature conductance derived from the transmission spectra in (b) at -1 eV while the corresponding dashed lines show the corresponding linear fit to $\ln G = -\beta L + constant$, where $L$ is the separation between two Br atoms. **d** The LDOS with yellow color in the energy window from –1.5 eV to -0.5 eV for 'L-R2' 'L-R3' and 'L-R4' separately at the isosurface 0.00008.

HOMO-LUMO gaps of MAPbBr$_3$ clusters of different sizes. As the size of the clusters decreases, their HOMO-LUMO gaps increase to 2.5 eV (for 16Pb) and further to 3.91 eV (for 1Pb) due to the stronger quantum confinement effect. After the rupture of the gold wire, the initial gap width is known to be a snap-back distance of about 5 Å. Since this corresponds to the distance between two neighboring Br atoms (around 5 Å), these two Br atoms are most likely to be connected to the gold atoms at the beginning in this sliding process. As for the sliding direction, the gold electrode could slide from the top of one Br atom to its adjacent Br atom along the horizontal and diagonal directions (shown as green and red arrows in Supplementary Fig. 30). The corresponding total energies upon sliding along one unit cell are shown in Supplementary Fig. 30c. Compared with displacement along the green 'horizontal' direction, the energy barrier is much higher in the red 'diagonal' direction due to the existence of CH$_3$NH$_3^+$ in the cavity. This demonstrates that there is a low-energy channel for sliding along the 'horizontal' direction, whereas the 'diagonal' direction contains a high-energy barrier and is less likely to happen in a real experiment. In order to further analyze the possible binding sites of the gold electrodes during the sliding process, we use the spectral clustering algorithm to give comprehensive and detailed classifications of the individual conductance-distance traces. The original conductance-distance traces can mainly be divided into five categories (as shown in Supplementary Figs. 20–22 and Note 5). The results show that although the three-step plateaus do not always appear simultaneously, the classified conductance plateaus display similar conductance features, i.e. the similar conductance values and displacement distributions, further confirming that the gold electrodes are more likely to slide along the 'horizontal' direction rather than the 'diagonal' direction. Therefore, the sliding along the 'horizontal' direction is adopted here to understand what we observed experimentally.

In the current study, the fully relaxed 12Pb MAPbBr$_3$ cluster is connected to two gold electrodes through two Br atoms as shown in Fig. 1f, where the Br atom labeled by 'L' is attached to the left gold electrode, the right gold electrode is attached successively to Br atoms labeled by 'R1', 'R2', 'R3', and 'R4' to model a sliding process. The corresponding transmission spectra are plotted in Fig. 4b. When the Fermi energy lies within the HOMO-LUMO gap, charge transfer occurs through the junction via off-resonant tunneling and the tunneling probability decays exponentially with L. Therefore, we fit the room-temperature conductance ($E_F = -1.0$ eV) to an exponential function, which led to an attenuation factor of $\beta = 0.54$ Å$^{-1}$ was obtained, which is consistent with our measured value of 0.57 Å$^{-1}$, as shown in Fig. 4c. More interestingly, in agreement with our experiments, when the right electrode is moved from 'R3' to the furthest distance 'R4', we obtain a much higher conductance compared with the shorter path 'R3'. This increase is also reflected in the qualitative behavior of the local density of states (LDOS) for 'L-R2', 'L-R3', and 'L-R4' (Fig. 4d). In contrast with R3, the weights of LDOS extend almost continuously between the left electrode and R4. This increase at the most distant electrode separation is also found in 8Pb, 10Pb (obtained by removing two Pb units based on the 2 × 2 × 3 12Pb MAPbBr$_3$), 16Pb and 18Pb MAPbBr$_3$ clusters (Supplementary Figs. 26–31 and Supplementary Figs. 36–37). Other possible connectivities for 16Pb MAPbBr$_3$ cluster are also explored (Supplementary Fig. 27–28). We find that this jump behavior is generic although different $\beta$ factors are observed (0.72 and 1.2 Å$^{-1}$ separately), and the latter connectivity is less likely to appear in the experiments due to the higher energy barrier.

The influence of ligand (oleic acid and octylamine) on the transmission functions is investigated by considering the ligands staying close to the cluster or bridging the gold electrode and perovskite cluster (see Supplementary Figs. 32 and 33). Our results reveal that the effect of ligand is negligible due to the weak coupling between ligands and cluster or gold electrode. We also carry out DFT calculations for MAPbCl$_3$ and MAPbI$_3$ QDs. Our results show that the three halide perovskite QDs possess similar charge transport features (see Supplementary Figs. 34 and 35). However, as mentioned above, in a real experiment they are not expected to form junctions, because of the weaker Au–Cl bond and the poorer stability of crystal structure for MAPbI$_3$ QDs[24–26]. Two new left binding sites (L$^a$ and L$^b$) are also considered in our calculations. As shown in Supplementary Figs. 38 and 39, we find the conductance evolution follows the same trend as the binding sites of L, i.e. it first decays exponentially and then jumps at the end. However, the magnitude of conductances is much smaller in these two cases due to the higher barrier caused by the larger Br–Br distance.

## Discussion

In summary, we have presented an experimental and theoretical investigation of room-temperature QI effects in the electron transport through single perovskite QD junctions, using a combination of DFT and the MCBJ technique. Three distinct conductance features are observed from the conductance measurements of perovskite QDs with Br, while the QDs with I and Cl show no significant features. The analysis of conductance trends with displacement reveals that the multiple conductance features are derived from the sliding of gold electrodes between the adjacent Br atoms in different unit cells. Counterintuitively, we also observe a distinct conductance 'jump' at the end of individual conductance traces, which is direct evidence of the room-temperature QI effects. This work offers an insight into QI effects in perovskite materials at the single-unit-cell level and also provides an opportunity to explore a strategy for optimizing electron transport in perovskite QDs electronic and optoelectronic devices.

## Method

**Synthesis of MAPbX$_3$ (MA = CH$_3$NH$_3^+$, X = I$^-$, Br$^-$, Cl$^-$, a mixture of Br$^-$ and Cl$^-$) perovskite QDs.** Perovskite QDs in this paper are synthesized according to published papers[21]. Typically, 0.2 mmol PbBr$_2$ (or PbCl$_2$ for MAPbCl$_3$, and PbI$_2$ for MAPbI$_3$) and 0.16 mmol MABr (or MACl for MAPbCl$_3$ and MAPbBr$_{2.15}$Cl$_{0.85}$, MAI for MAPbI$_3$ and MAPbBr$_{2.15}$I$_{0.85}$) is dissolved in 5 mL DMF. 0.5 mL oleic acid and 20 μL n-octylamine are added to obtain a stable precursor solution. The 1 mL precursor solution is rapidly injected into 5 mL toluene under the stirring with 800 rpm. In the stirring process, strong green PL emission from MAPbBr$_3$ QDs can be observed under normal room light without using additional excitation light source. Then the precursor solution is centrifuged at 5000 or 10,000 rpm for 10 min to discard the precipitates.

**Characterization.** Transmission electron microscopy (TEM) and high-resolution TEM (HRTEM): a drop of diluted QDs solution is spread onto an ultrathin carbon film-coated copper grid and is further dried by gentle N$_2$ blowing. Transmission electron microscopy (TEM, JEOL JEM-2000EX) with fast operation at 200 kV is employed to obtain TEM or HRTEM images before damaging perovskite QDs.

**The single-molecule conductance measurement.** The single-molecule conductance measurements of perovskite QDs are carried out by using the MCBJ technique in a solution containing 0.365 mg mL$^{-1}$ QDs at room temperature[22]. During the process of MCBJ measurement, the substrate is first fixed by two counter supports at both sides, and then a pushing rod driven by a stepping motor and a piezo stack is employed to bend the substrate in the middle, resulting in the breakage of the notched gold wire at the horizontal direction to form a nano-gap. The fractured gold wire will capture a single perovskite QD with halogen anchors and forms the Au-QD-Au junction. Owing to the elasticity of the stainless-steel substrate, the gold wire connects again during the returning process of the pushing rod. After repeating this process thousands of times, the individual conductance-

distance traces of the single-QD junctions can be collected and further analyze the most probable conductance of the junctions.

**DFT theoretical calculation.** Using the DFT code SIESTA [29] geometrical optimizations were carried out until all the forces were less than 0.05 eV Å$^{-1}$. A generalized gradient approximation functional (GGA), a double-$\zeta$ basis for Au, a double-$\zeta$ polarized basis for other elements and a real grid cutoff energy of 150 Ry were employed[29,30]. A scalar relativistic norm-conserving pseudopotential is used to describe Pb. To compute their electrical conductance, the molecules are each placed between pyramidal gold electrodes. After relaxation, the optimized separation between contact halogen atoms (Cl, Br, I) and apex gold atom was found to be 2.66 Å, 2.76 Å and 2.88 Å, respectively. From the converged DFT calculation of each structure, the mean field Hamiltonian and overlap matrix are extracted, which are utilized to calculate the transmission coefficient $T(E)$ using the Gollum code [30], via the expression

$$T(E) = Tr\left[\Gamma_L(E)G_r(E)\Gamma_R(E)G_r^\dagger(E)\right]. \tag{1}$$

In this equation, $\Gamma_{L,R}(E) = i(\Sigma_{L,R}(E) - \Sigma_{L,R}^\dagger(E))/2$. $\Gamma_{L,R}$ describes the level broadening due to the interaction between left (right) electrodes and the scattering region. $\Sigma_{L,R}(E)$ are the self-energies. $G_r = (ES - H - \Sigma_L - \Sigma_R)^{-1}$ is the retarded Green's function, where $S$ and $H$ are the Hamiltonian and overlap matrix, respectively. The room-temperature conductance is obtained by the following formula: $G = G_0 \int_{-\infty}^{+\infty} dE T(E)(-\frac{\partial f(E)}{\partial E})$, where $G_0 = 2e^2 h^{-1}$ is the conductance quantum; $h$ is the Planck's constant; $e$ is the charge of a proton; $f(E) = (1 + \exp((E - E_F)/k_B T))^{-1}$ is the Fermi–Dirac probability distribution function, $E_F$ is the Fermi energy.

**Reporting summary.** Further information on research design is available in the Nature Research Reporting Summary linked to this article.

## Data availability
The source data underlying Fig. 1e, Fig. 1g, Fig. 2b-d, Fig. 3a-f and Fig. 4a-c are provided as a Source Data file. All other data are available from the corresponding author upon reasonable requests.

## Code availability
The source code of the algorithms are available for research uses at https://github.com/zhenghaining121/QDs-codes. Computational data used to arrive at the conclusions presented in the manuscript are available upon reasonable request.

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

## Acknowledgements
This work was supported by the National Key R&D Program of China (2017YFA0204902, 2014DFE60170, 2018YFB1500105), the National Natural Science Foundation of China (Nos. 21673195, 21503179, 21490573, 61674084, 61874167), the Open Fund of the Key Laboratory of Optical Information Science & Technology (Nankai University) of China, the Fundamental Research Funds for the Central Universities of China (63181321, 63191414, 96173224), and the 111 Project (B16027), the Tianjin Natural Science Foundation (17JCYBJC41400), FET Open project 767187—QuIET, the EU project BAC-TO-FUEL and the UK EPSRC projects EP/N017188/1, EP/M014452/1.

## Author contributions
W.H., C.L., and Y.L. designed the experiments and co-supervised the project. H.Z. and S. H. wrote the manuscript with inputs from all authors. H.Z., Z.T., F.J., J.Z., J.P., W.H., Q. L., and J.L. carried out the break junction experiments and analyzed the data. L.L. provided the spectral clustering algorithm for the classifications of the individual conductance-distance traces. C.X., X.Z., Y.L., X.Z., and Y.Z. were responsible for molecular synthesis and characterization. J.S., L.Z., and Y.Y. built the electrical measurement instrument and wrote the software to control the break junction setup. C.L., S.H. and Q. W. developed the underlying theoretical concepts. All authors conceived the work and discussed the experiments.

## Competing interests
The authors declare no competing interests.
