## [Peer Review File · Nature Communications]

Reviewers' comments:

Reviewer #1 (Remarks to the Author):

The manuscript by Zheng et al. presents a single molecular conductance study of Perovskite quantum dot based on MCBJ technique. The authors observed interesting step-wise behavior of single molecular conductance plateau. In addition to that, the authors also observed a jump at the end of each trace. This was attributed to a switch from destructive quantum interference to constructive interference. Overall, I found the work is novel and significant. I have several concerns that I hope the authors could address before publication.

1. My most significant concern is that the QDs are not monodisperse (from the characterizations in the SI). Even if it is monodisperse, the measured single molecule (one single QD) is not a single pure molecule with well-defined chemical structure, and this is usually not good for single molecular conductance measurement. Several issues regarding this point:

a. The contact sites could be very complicated in this case. As given in Fig. 2a, a 12 Pb MAPbBr₃ is used as an example. There are 6+10+10+6=32 contact sites (or the Br atoms) that are exposed, and potentially they can bind with Au electrode. The possible binding configuration could be $32 \times 31 = 992$. However, in the current manuscript, the authors only treat the whole cluster as one-dimensional array. For example, if the other Au electrode slides through along the diagonal direction of the cluster (see picture below). Will it bring different behavior in the conductance-distance trace? I could imagine that among the 992 configurations, there might be some similar cases of destructive/constructive interference, thus giving several plateaus then one jump, but do they give the same conductance value?

b. As the authors mentioned in the text and figures, in most cases they observed three plateaus and then one jump. This observation is unexpected to me. Why three plateaus are most common? Is it because most QDs are $2 \times 2 \times 3$, which is 12 Pb MAPbBr₃ as shown in Fig 2a? But the QDs are not monodisperse. Even if most QDs are 12 Pb MAPbBr₃, the sliding of the other electrode could go along all directions, e.g. giving two

plateaus and then one jump, or several plateaus without a jump (see sliding directions above). The most common plateau is 3-step plateau. This is quite confusing to me.

c. This may also cause some issues on Fig. 3c, the displacement distribution, because not all the trace will follow the same direction across the QDs.

d. In sum, I suggest the authors to carry out more analysis on their data. For example, the authors could analyze their traces and figure out the percentage of three plateaus/two plateaus/one plateaus, with or without the final jump. And then analyze all the traces in each category, to get the conductance value and displacement distribution. This could help resolving the issues of arbitrary contact sites.

2. The authors should provide the chemical structure of 1 MAPbBr₃ in Fig. 1. This helps people see clearly what could be the contact sites. The ball-stick model is too small to visualize. Also, when mentioning MAPbBr₃, authors should explain what is MA here, because people in the field of single molecular electronics are not familiar with Perovskite.

3. Panel d seems to contradict panel f. In panel d, $2 < 3$, but in panel f, $2 > 3$. There might be some mislabeling issues.

4. The explanation of the sign dependence in quantum interference is a bit unclear. The text is too long and hard to understand. Authors may consider use one equation to explain it concisely (see *Acc. Chem. Res.* 2012, 45, 9, 1612-1621).

5. What is the dash line in Fig. 2d and Fig. 3e? Does it indicate the slope of the jump? I do not see any discussions about this in this current manuscript.

Reviewer #2 (Remarks to the Author):

In this manuscript, Zheng et al. reported a combined experimental and theoretical work on the quantum interference effect by investigating the conductance, and three distinct conductance features are observed from Br contained nanocrystals (NCs), while in Cl or I based samples, there are no such features. Overall, the experimental finding of the conductance features in Br contained perovskite NCs is very interesting, however, based on my opinion, the proposed explanation or origin of these features is quite vague, therefore, I cannot recommend it for publication in its current form, the details are listed below:

1. The binding detail of between gold and perovskite NC (or quantum dot QD) is questionable, clearly, n-octylamine and oleic acid are used to stabilize the NC, such that these organic ligands occupy the outer space of perovskite NCs, although the formation of Au-Br bonds are independently supported by Raman spectra and the conductance of oleic acid and octylamine are provided in SI, this Au-Br could be from the surface Br ions along with these organic ligands, or the migration of Br ions;
2. I also have concerns about the QD structures in the computation, as mentioned above, the information of organic ligands is completely missing, can these structures represent the ones in the experiments? Intuitively, I think the organic ligands have impacts on the interaction between gold and QDs, also the electronic structure of QDs.
3. The authors provided statistical data on I, Cl, Br and Br/Cl samples, but the repeatability of the data on a specific sample, for example Br is not well supported, it would be interesting to see the experimental data on a few different pure Br based QD samples;
4. The authors show the theoretical results agrees well with the experimental, however, a comparison between the theoretical results also on Cl and I based QDs is needed to distinguish the Br QDs to the Cl and I based ones.

Reviewer #3 (Remarks to the Author):

This manuscript has demonstrates experimental and theoretical investigation of room temperature QI effects in the electron transport through single perovskite QD junctions. Three distinct conductance features are observed from the conductance measurements of perovskite QDs with Br, while the QDs with I and Cl show no significant features. The multiple conductance features are derived from the sliding of gold electrodes between the adjacent Br atoms in different unit cells. A distinct conductance jump at the end of individual conductance traces, which is claimed as room-temperature QI effects. Basically this is interesting work and could possibly published in Nature Communications only after the following points are considered.

1. The distinct conductance jump at the end of individual conductance traces is very interesting and owing to the junction switch from "L-R3" to "L-R4". However, this is ideal case only for three lattices. From the TEM images of Fig. S2 and S3, the QDs are not uniform enough. To support the authors' conclusion, more cases with different numbers of lattices should be included. Then the conductance jumps may appear after two or four (and so on) conductance plateaus.
2. Three exposed halogen atoms on the corner lattice had the opportunity to interact with the gold electrodes. One of them was labelled as "L" in this work. The authors only assumed and calculated one ideal case. How about the other two cases for the "L" atom in two orthogonal directions? The conclusion from just one special case is not convinced enough since the authors cannot precisely control the junction between the QD and the Au electrodes.
3. The motivation for using $\text{MAPbBr}_{3-x}\text{Cl}_x$ QDs should be stated. And why not $\text{MAPbBr}_{3-x}\text{I}_x$? Moreover, the exact values of $3-x$ and x should be given in the formula.
4. The authors claimed the junction is based on Au-Br interactions. Therefore, why no conductance plateaus can be observed for PbBr_2 ? I doubt the control experiment was not reasonably carried out.

5. I do not believe Au-I bond is strong enough to break the stable crystal structure of MAPbI₃ with the pulling process of the electrodes. This explanation needs further support.

Point-to-point reply to reviewers' comments

Reviewer: 1

Comments:

The manuscript by Zheng et al. presents a single molecular conductance study of Perovskite quantum dot based on MCBJ technique. The authors observed interesting step-wise behavior of single molecular conductance plateau. In addition to that, the authors also observed a jump at the end of each trace. This was attributed to a switch from destructive quantum interference to constructive interference. Overall, I found the work is novel and significant. I have several concerns that I hope the authors could address before publication.

(1) My most significant concern is that the QDs are not monodisperse (from the characterizations in the SI). Even if it is monodisperse, the measured single molecule (one single QD) is not a single pure molecule with well-defined chemical structure, and this is usually not good for single molecular conductance measurement. Several issues regarding this point:

a. The contact sites could be very complicated in this case. As given in Fig. 2a, a 12 Pb MAPbBr₃ is used as an example. There are $6+10+10+6=32$ contact sites (or the Br atoms) that are exposed, and potentially they can bind with Au electrode. The possible binding configuration could be $32 \times 31 = 992$. However, in the current manuscript, the authors only treat the whole cluster as one-dimensional array. For example, if the other Au electrode slides through along the diagonal direction of the cluster (see picture below). Will it bring different behavior in the conductance-distance trace? I could imagine that among the 992 configurations, there might be some similar cases of destructive/constructive interference, thus giving several plateaus then one jump, but do they give the same conductance value?

Response: Thanks very much for your suggestion. Yes, for 12Pb MAPbBr₃, we agree that there could be 992 possible binding configurations in principle, although after taking symmetry into consideration, this number would be reduced, because the CH₃NH₃PbBr₃ perovskite-type unit cell is a regular octahedral structure which is centrosymmetric (as shown in Fig 1). When the gold electrode slides along the ‘horizontal’ directions, all the cases shown as green arrows (in Fig. 1 below) will yield the same experimental results. In addition, in real experiments, after the rupture of the gold wire, the initial gap width is known to be a snap-back distance of about 5 Å. Since this corresponds to the distance between two neighbouring Br atoms (around 5 Å), these two Br atoms are most likely to be connected to the gold electrodes at the beginning of this pulling process. Therefore, the contact points are not completely arbitrary at the beginning of the junction formation and the possible binding configurations would be far less than 992.

Fig. 1 | The schematic of the pulling process of gold electrodes. The green arrows represent the ‘horizontal’ directions. The red arrows represent the ‘diagonal’ directions.

Indeed, we cannot rule out the possibility that the gold electrodes can also slide along the ‘diagonal’ directions (shown as red arrows in Fig. 1). However, taking the $2 \times 3 \times 3$ 18Pb MAPbBr₃ as an example, the gold electrodes only have 2 possibilities that slide along the ‘diagonal’ directions, while the ‘horizontal’ directions would be 10 possibilities, which is five times more than the former. Therefore, the gold electrodes have more chance to slide along the ‘horizontal’ directions. In addition, we also analyze the energies of sliding along these two directions. The corresponding total energies upon sliding along one unit cell are shown in Fig. S30c. Compared with displacement along the green ‘horizontal’ direction, the energy barrier is much higher in the red ‘diagonal’ direction, due to the existence of CH₃NH₃⁺ in the cavity. In addition, the distance between two adjacent Br atoms along ‘diagonal’ direction (around 0.8 nm) is much larger than the experimental value (the plateau length difference is about 0.5 nm see Fig. 3c in manuscript). This demonstrates that there is a low-energy channel for sliding along the ‘horizontal’ direction, whereas the ‘diagonal’ direction contains a high-energy barrier and is less likely to happen in a real experiment.

To enhance our understanding of the experiment, we added Fig. S30 to Supplementary Information (SI) and the following sentences to the manuscript:

“After the rupture of the gold wire, the initial gap width is known to be a snap-back distance of about 5 Å. Since this corresponds to the distance between two neighbouring Br atoms (around 5 Å), these two Br atoms are most likely to be connected to the gold atoms at the beginning in this pulling process. As for the sliding direction, the gold electrode could slide from the top of one Br atom to its adjacent Br atom along the horizontal and diagonal directions (shown as green and red arrows in Fig. S30). The corresponding total energies upon sliding along one unit cell are shown in Fig. S30c. Compared with displacement along the green ‘horizontal’ direction, the energy barrier is much higher in the red ‘diagonal’ direction due to the existence of CH₃NH₃⁺ in the

cavity. This demonstrates that there is a low-energy channel for sliding along the ‘horizontal’ direction, whereas the ‘diagonal’ direction contains a high-energy barrier and is less likely to happen in a real experiment. Therefore, the sliding along the ‘horizontal’ direction is adopted here to understand what we observed experimentally.”

Fig. S30 | Energy landscapes versus the displacement of the gold electrode along two different directions. a, Side view of 2x2x3 12Pb MAPbBr₃ with one gold electrode. Two sliding directions are indicated by green and red arrows separately. **b,** Top view. The gold electrode is not displayed for clarity. **c,** Profiles of the total energies when moving the gold lead along these two directions with a fixed height (2.76 Å) from the line formed by the two Br atoms. L is the distance between two Br atoms along the sliding direction.”

Nevertheless, for completeness, we investigated theoretically the conductance evolution along the ‘diagonal’ direction using a larger 3x3x2 18Pb MAPbBr₃ cluster (MA₃₉Pb₁₈Br₇₅), as shown in Fig. S31. The diagonal sliding indicated by the red arrow and the junction conformations during this process are shown in Fig. S31a. The corresponding transmission functions and room-temperature conductances are

presented in Fig. S31b and S31c. In common with the conductance evolution in the ‘horizontal’ direction, as the electrode slides along the ‘diagonal’ direction, the conductance initially decays exponentially, but just before rupture, there are two possible contact Br sites as presented in L-R4^a and L-R4^b. For L-R4^b, we again find a conductance jump at the end of the pulling, while there is no jump for L-R4^a. Furthermore the ‘diagonal’ direction yields a higher β factor of 0.87, compared with 0.58 along the ‘horizontal’ direction. The larger β factor is a result of the weak coupling between the discrete Pb unit cells in the ‘diagonal’ direction.

To clarify this point, we added Fig. S31 to the SI:

Fig. S31 | The DFT theoretical calculation of 18Pb MAPbBr₃ cluster when sliding along the diagonal direction. a, The top view of 18Pb cluster, the diagonal sliding direction is indicated by the red arrow. The conformations for 18Pb MAPbBr₃ cluster embedded in two gold electrodes with 5 different connectivity L-R1, L-R2, L-R3, L-R4^a and L-R4^b. **b-c**, The corresponding transmission spectra and the room temperature conductance with the $E_F = -1$ eV as indicated by the black dashed line in **b**, the $\beta \approx -0.87$.”

b. As the authors mentioned in the text and figures, in most cases they observed three plateaus and then one jump. This observation is unexpected to me. Why three plateaus are most common? Is it because most QDs are $2 \times 2 \times 3$, which is 12 Pb MAPbBr₃ as shown in Fig 2a? But the QDs are not monodisperse. Even if most QDs are 12 Pb MAPbBr₃, the sliding of the other electrode could go along all directions, e.g. giving two plateaus and then one jump, or several plateaus without a jump (see sliding directions above). The most common plateau is 3-step plateau. This is quite confusing to me.

Response: Thank you for your questions. Indeed, from the HRTEM image in Fig. S2 and S4, the QDs are not monodisperse. In other words, these QDs have various sizes and diameters. In fact, it is difficult to guarantee the QDs are monodisperse and all have similar diameters in the synthesis process even though the synthetic experiments were carried out in the glovebox and controlled the water and oxygen in the environment strictly. (Dong, Y., *et al. ACS Nano* **9**, 4533-4542 (2015); Andrey, R., *et al. Adv. Sci.* **2**, 1500194, (2015); Zhong, H., *et al. Chem. Mater.* **298**, 3793-3799 (2017)) However, from the high-resolution transmission electron microscopy (HRTEM) images of the MAPbBr₃ and MAPbBr_{3-x}Cl_x QDs shown in Fig. S2 and S4, although these two kinds of QDs display a significant difference in diameter distributions (3.75 ± 1.39 nm and 6.32 ± 1.63 nm, respectively), as shown in Fig. 3b, there is little discernible difference in their conductance features (three clear conductance plateaus and almost same conductance values), indicating that the diameters and sizes of the QDs have no distinct impact on their electrical properties. This is because the three conductance plateaus

come from the perovskite unit cells through Au-Br interaction rather than the entire QDs. From Fig. 3c of the manuscript, the average displacement differences of the three plateaus (~ 0.5 nm) match well with the adjacent lattice distances of Br (~ 0.5 nm) (Pérez-Prieto, J., *et al. J. Am. Chem. Soc.* **136**, 850-853 (2014); Zhao Y., *et al. Chem. Commun.* **51**, 7820-7823, (2015)) rather than the diameters of the QDs (3.75 nm or 6.32 nm). In addition, we also carried out control experiments of all the ligands and raw materials (Fig. S10) and no obvious conductance signals could be observed except PbBr₂, indicating that only Br atoms can provide the binding sites of gold electrodes and the ligands outside the QDs have no impact on the charge transport of QDs. Therefore, as discussed above, it is reasonable to conclude that even though QDs are not monodisperse, this has no impact on their charge transport properties.

To further verify the above conclusions, we carried out the MCBJ measurements of MAPbBr₃ QDs obtained from the centrifugal speeds of 5000 and 10000 rpm, respectively (as shown in Fig. S3). The HRTEM images suggest that the QDs obtained from the centrifugal speeds of 10000 rpm display the similar average diameters (6.34 nm) to the centrifugal speeds of 5000 rpm (6.35 nm), while the former has the smaller standard deviation (1.35 nm) than the latter (2.27 nm), further proving that although the QDs are not monodisperse, the diameters and distributions of QDs do not affect the MCBJ experimental results.

To give a more detailed demonstration, we add the following sentences and the corresponding figures to **SI**:

Fig. S3 | The image of transmission electron microscopy for MAPbBr₃ perovskite QDs with different centrifugal speeds. a, The HRTEM image for MAPbBr₃ perovskite QDs with the centrifugal speed of 5000 rpm. **b,** The HRTEM image for MAPbBr₃ perovskite QDs with the centrifugal speed of 10000 rpm.

S3.9 MCBJ measurements of MAPbBr₃ QDs with different centrifugal speeds

In order to prove that the sizes and diameters of QDs have no impact on their electrical properties, we carried out the MCBJ measurements using the perovskite quantum dots obtained with the centrifugal speeds of 5000 rpm and 10000 rpm (as shown in Fig. S3). The MAPbBr₃ QDs centrifuged with different centrifugal speeds express similar conductance values located at $10^{-1.54}$, $10^{-2.80}$ and $10^{-4.32}$ for 5000 rpm, $10^{-1.43}$, $10^{-2.72}$ and $10^{-4.28}$ for 10000 rpm, respectively, and the difference of adjacent statistical lengths matches well with the adjacent lattice distance of Br, which prove that the conductance plateaus we measured originate from the perovskite crystal cells rather than the entire perovskite QDs.

Fig. S17 | The MCBJ measurements of MAPbBr₃ QDs with different centrifugal speeds. **a.** 1D Conductance histogram constructs without data selection for MAPbBr₃ with the centrifugal speed of 5000 rpm. The conductance-distance traces are recorded ~2500 traces. **b.** All-data-points 2D conductance versus relative distance (Δz) histogram for MAPbBr₃ with the centrifugal speed of 5000 rpm. **c.** The displacement distributions of three plateaus for MAPbBr₃ with the centrifugal speed of 5000 rpm. **d.** 1D Conductance histogram constructs without data selection for MAPbBr₃ with the centrifugal speed of 10000 rpm. ~2500 conductance-distance traces are recorded. **e.** All-data-points 2D conductance versus relative distance (Δz) histogram for MAPbBr₃ with the centrifugal speed of 10000 rpm. **f.** The displacement distributions of three plateaus for MAPbBr₃ with the centrifugal speed of 10000 rpm.”

In addition, we also added the following sentences in the manuscript:

“In addition, we also carry out the MCBJ measurements using the MAPbBr₃ QDs with the average diameters of 6.34 nm and 3.75nm, which are obtained from the centrifugal speeds of 10000 rpm and 5000 rpm, respectively (as shown in Fig. S3 and Fig. S17). The experimental results show that the QDs with different diameters show similar conductance features, indicating that the conductance plateaus we measured originate from the perovskite crystal cells rather than the entire perovskite QDs.”

Regarding your concerns of the multiple possible binding sites during the sliding process of the gold electrodes, as we explained in the first response, the theoretical results show that the gold electrodes are more likely to slide along the ‘horizontal’ direction rather than the ‘diagonal’ direction due to the much lower the energy barrier. In addition, the distance between two adjacent Br atoms along the ‘horizontal’ direction is in accordance with the experimental value.

It is possible that the gold electrode could go through one plateau, two plateaus, three plateaus and even more plateaus. The reason that more than three plateaus are not shown is due to the fact that the forth plateaus may be lower than the sensitivity limits of our set-ups. Furthermore, in order to analyze whether the 3-step plateaus are the most common cases, we use the spectral clustering algorithm to give comprehensive and detailed classifications of the individual conductance-distance traces. The original conductance-distance traces can mainly be divided into five categories. As for MAPbBr₃ QDs: 15.5% of traces with three successive conductance plateaus, 35.0% of traces with the highest conductance (HC) and middle conductance (MC), 31.3% of traces with the HC and the lowest conductance (LC), 9.9% of traces with MC and LC, and 8.3% of traces with only MC. As for the MAPbBr_{2.15}Cl_{0.85} QDs: 16.2% of traces with three successive conductance plateaus, 44.0% of traces with HC and MC, 21.3% of traces with the HC and LC, 8.7% of traces with MC and LC, and 9.8% of traces with only MC. From these classification results, it is demonstrated that the 3-step plateaus are indeed not the most common causes during the pulling process. When the experimental environment and instrument condition are excellent, we can even observe four distinct conductance plateaus with conductance “jump” at the end of the conductance traces (as shown in Fig. S22). Even so, these conductance plateaus display similar conductance features, ie the similar conductance values and displacement distributions which are also in accordance with the adjacent distances of Br atoms, further confirming that the gold electrodes are more likely to slide along the ‘horizontal’ direction rather than the ‘diagonal’ direction.

To clarify this point, we have added the following sentences in the manuscript and **SI**:

“In order to further analyze the possible binding sites of the gold electrodes during the pulling process, we use the spectral clustering algorithm to give comprehensive and detailed classifications of the individual conductance-distance traces. The original conductance-distance traces can mainly be divided into five categories (as shown in Fig. S20 and S21). The results show that although the three-step plateaus do not always appear simultaneously, the classified conductance plateaus display similar conductance features, ie the similar conductance values and displacement distributions, further confirming that the gold electrodes are more likely to slide along the ‘horizontal’ direction rather than the ‘diagonal’ direction.”

“S3.11 The classification results of MAPbBr₃ and MAPbBr_{2.15}Cl_{0.85} QDs using spectral clustering algorithms

The spectral clustering algorithm is a state of art clustering technique which provides a partition of data and assigns similar data traces into clusters. Here we firstly review the spectral clustering algorithm according to the Ng *et al.* on the 1D conductance histograms.

Given histogram data $H = \{h_1, h_2, \dots, h_M\}$ in R^N (dividing the conductance axis to discrete N bins) that we want to cluster into K clusters:

1. Form the affinity matrix $A \in R^{M \times M}$ defined by $A_{ij} = C_{ij} + 1$ if $i \neq j$, and $A_{ii} = 0$.
2. Define D to be the diagonal matrix whose (i, j) -element is the sum of A 's i -th row, and construct the matrix $L = D^{-1/2} A D^{-1/2} - I$.
3. Find x_1, x_2, \dots, x_K , the K largest eigenvectors of L , and form the matrix $X = [x_1, x_2, \dots, x_K]$ belong to $R^{M \times K}$ by stacking the eigenvectors in columns.
4. Treating each row of X as a point in R^K , cluster them into K clusters via K-means++.
5. Finally, assign the original points h_i to cluster j if and only if row i of the matrix X was assigned to cluster j .

Here we construct the affinity matrix A specified different from the usual one (the Gaussian (aka RBF) kernel), the other steps are almost the same as described in Ng *et*

a). Here we define the C_{ij} as the cross-correlation between histogram h_i and h_j as follow:

$$C_{ij} = \frac{\langle [h_i - \langle h_i \rangle][h_j - \langle h_j \rangle] \rangle}{\sqrt{\langle [h_i - \langle h_i \rangle]^2 \rangle \langle [h_j - \langle h_j \rangle]^2 \rangle}} \quad (1)$$

where $\langle h_i \rangle$ represents the average value of histogram h_i , the values of C range from $[-1, 1]$, so we add one to make the elements of affinity matrix A nonnegative to meet the spectral clustering requirements. h_i is the conductance histogram for the i -th individual trace, M is the number of conductance traces, N is the number of the histogram bins.

25. Ng, A.Y., Jordan, M.I. & Weiss, Y. On spectral clustering: Analysis and an algorithm. in *Advances in neural information processing systems* 849-856 (2002).

Fig. S20 | The classification results of MAPbBr₃ QDs using spectral clustering

algorithms. a, 15.5% of traces with three successive conductance plateaus. b, 35.0% of traces with the highest conductance (HC) and middle conductance (MC). c, 31.3% of traces with the HC and the lowest conductance (LC). d, 9.9% of traces with MC and LC. e, 8.3% of traces with only MC.

Fig. S21 | The classification results of MAPbBr_{2.15}Cl_{0.85} QDs using spectral clustering algorithms. a, 16.2% of traces with three successive conductance plateaus. b, 44.0% of traces with the highest conductance (HC) and middle conductance (MC). c, 21.3% of traces with the HC and the lowest conductance (LC). d, 8.7% of traces with MC and LC. e, 9.8% of traces with only MC.

Fig. S22 | The classification results of MAPbBr₃ QDs with four conductance plateaus using spectral clustering algorithms. a, 1D Conductance histogram constructs without data selection for MAPbBr₃ QDs. **b,** All-data-points 2D conductance versus relative distance (Δz) histogram for MAPbBr₃ QDs. **c,** The displacement distributions of three plateaus for MAPbBr₃ QDs. **d,** 2D relative conductance (G) versus relative displacement (Δz) histogram of the “jump curves” for MAPbBr₃ QDs.”

c. This may also cause some issues on Fig. 3c, the displacement distribution, because not all the trace will follow the same direction across the QDs.

Response: Thank you for this nice suggestion. As mentioned in the above response, we have given a comprehensive classification of the traces using spectral clustering algorithms and obtained the relative displacement distributions. From these classification results, it is clear that although the three conductance plateaus do not appear at the same time in most cases, each plateau in different categories shows the similar displacement distributions, further confirming that the gold electrodes are more likely to slide along the “horizontal” directions rather than the “diagonal” directions.

d. In sum, I suggest the authors to carry out more analysis on their data. For example, the authors could analyze their traces and figure out the percentage of three plateaus/two plateaus/one plateaus, with or without the final jump. And then analyze all the traces in each category, to get the conductance value and displacement distribution. This could help resolving the issues of arbitrary contact sites.

Response: Thank you for this suggestion. As mentioned in the above response, we apply the spectral clustering algorithms to classify all the conductance traces in five categories and get the relative conductance values and displacement distributions in each category. In order to correlate each category to the possible binding sites, we

expanded the theoretical models with different numbers of lattices (8Pb ($\text{MA}_{20}\text{Pb}_8\text{Br}_{36}$), 10Pb ($\text{MA}_{24}\text{Pb}_{10}\text{Br}_{44}$), 12Pb ($\text{MA}_{28}\text{Pb}_{12}\text{Br}_{52}$) and 16Pb ($\text{MA}_{36}\text{Pb}_{16}\text{Br}_{68}$)) and more binding sites (L^a and L^b). More Pb units would be too computationally expensive since 18Pb MAPbBr_3 has 405 atoms already. Our theoretical results of the Br-Br distances and conductances were summarized in Table S2 (shown as follows). It is demonstrated that the conductances of clusters with different numbers of lattices display similar features (the conductance first decays exponentially and then jump at end), while the attenuation factors β of these different cases vary greatly (from $0.54 / \text{\AA}$ to $1.2 / \text{\AA}$). Comparing these theoretical conductances and Br-Br distances with our classified experimental results, it is clear that only the case that the gold electrodes connect to the closest Br atoms after rupture and sliding along the ‘horizontal’ direction can well support our experimental results (such as Fig. 4c, S27 and S36). The number of clusters has no distinct impact on theoretical results.

To clarify this point, we added Table S2 and Fig. S30-S39 in SI and also added these sentences in manuscript:

“This increase at the most distant electrode separation is also found in 8Pb, 10Pb (obtained by removing two Pb units based on the $2 \times 2 \times 3$ 12Pb MAPbBr_3), 16Pb and 18Pb MAPbBr_3 clusters (Fig. S26-S31 and S36-S37).”

and,

“Two new left binding sites (L^a and L^b) were also considered in our calculations. As shown in Fig. S38 and Fig. S39, we find the conductance evolution follows the same trend as the binding sites of L , ie it first decays exponentially and then jumps at end. However, the magnitude is much smaller in these two cases due to the higher barrier caused by the larger Br-Br distance.”

Fig. 36 | DFT calculations of 3x3x2 18Pb MAPbBr₃ cluster. a, Conformations for 18Pb MAPbBr₃ cluster embedded in two gold electrodes with 4 different connectivity L-R1, L-R2, L-R3 and L-R4. **b,** The corresponding transmission functions and room temperature conductances with the $E_F = -1$ eV relative to that estimated by DFT which is indicated by the black dashed line in the left panel.

Fig. 37 | DFT calculations of 2x2x3 10Pb MAPbBr₃ cluster. a, Conformations for 10Pb MAPbBr₃ cluster embedded in two gold electrodes with 4 different connectivity L-R1, L-R2, L-R3 and L-R4. **b,** The corresponding transmission functions and room temperature conductances with the $E_F = -1$ eV relative to that estimated by DFT which is indicated by the black dashed line in the left panel.

Fig. 38 | DFT calculations of 2x2x3 12Pb MAPbBr₃ cluster with a new left contact Br atom 'L^a'. a, Conformations for 12Pb MAPbBr₃ cluster embedded in two gold electrodes with 4 different connectivity L^a-R1, L^a-R2, L^a-R3 and L^a-R4. b, The corresponding transmission functions and room temperature conductances with the $E_F = -1$ eV relative to that estimated by DFT which is indicated by the black dashed line in the left panel.

Fig. 39 | DFT calculations of 2x2x3 12Pb MAPbBr₃ cluster with the other new left contact Br atom 'L^b'. a, Conformations for 12Pb MAPbBr₃ cluster embedded in two gold electrodes with 4 different connectivity L^b-R1, L^b-R2, L^b-R3 and L^b-R4. b, The corresponding transmission functions and room temperature conductances with the E_F = -1 eV relative to that estimated by DFT which is indicated by the black dashed line in the left panel.

Table S2 | The summary of theoretical results

Br-Br distance (Å)						
Conductance (log(G/G ₀))	R1	R2	R3	R4	R5	β (Å ⁻¹)
8Pb	5.09	9.85	12.57			
(Fig. S26)	-2.61	-4.46	-2.69			
12Pb	4.77	9.55	15.40	18.47		0.54
(Fig. 4c)	-3.08	-3.81	-5.52	-3.90		
16Pb	4.71	9.41	15.43	20.06	24.21	0.54
(Fig. S27)	-3.11	-3.80	-5.31	-6.67	-5.17	
16Pb-cross	5.33	11.43	16.13	20.7		0.72

(Fig. S28)	-4.54	-6.64	-7.89	-6.44		
16Pb-cross (Fig. S29)	7.94	12.71	17.64	21.64		1.2
18Pb-diagonal (Fig. S31)	4.98	10.89	18.2	20.84	20.63	0.87
18Pb (Fig. S36)	4.71	9.38	15.39	18.28		0.55
10Pb (Fig. S37)	5.19	9.91	14.94	18.31		0.98
12Pb-L ^a (Fig. S38)	6.76	11.00	16.00	18.78		0.42
12Pb-L ^b (Fig. S39)	10.53	13.7	17.98	19.47		0.88

»

(2) The authors should provide the chemical structure of 1 MAPbBr₃ in Fig. 1. This helps people see clearly what could be the contact sites. The ball-stick model is too small to visualize. Also, when mentioning MAPbBr₃, authors should explain what is MA here, because people in the field of single molecular electronics are not familiar with Perovskite.

Response: Sorry for the misunderstanding. We have changed the ball-stick model in Fig. 1 to the chemical structure of 1 MAPbBr₃, indicated by the following:

«

Fig. 1 | The orthogonality of molecular orbitals and QI of 1Pb perovskite cluster.

a, Schematic of coherent tunneling across a molecule, where the HOMO and LUMO of a one-dimensional chain of 8 sites are plotted. b, Left panel: The chemical structure

of a relaxed 1Pb MAPbBr₃ (MA= CH₃NH₃⁺) cluster. Middle panel: its corresponding HOMO and LUMO.....”

In addition, the MA represents CH₃NH₃⁺, which has been explained in our previous version:

“Four types of organic-inorganic halide perovskite QDs MAPbX₃ (MA= CH₃NH₃⁺, X=I⁻, Br⁻, Cl⁻, a mixture of Br⁻ and Cl⁻) are synthesized with oleic acid and octylamine as ligands to enhance colloidal stability and suppress QD aggregation effects.”

In order to make it clearer, we also added the corresponding explanation in the abstract:

“Single-QD conductance measurements reveal that there are multiple distinguishable conductance peaks for the MAPbBr₃ and MAPbBr_{2.15}Cl_{0.85} QDs (MA= CH₃NH₃⁺), whose displacement distributions match the lattice constant of QDs, suggesting that the gold electrodes slide through different lattice sites of the QD via Au-halogen coupling.”

(3) Panel d seems to contradict panel f. In panel d, 2<3, but in panel f, 2>3. There might be some mislabeling issues.

Response: Thank you for noting this error. There is indeed a mislabeling issue. We have changed the Fig. 1 as follows:

Fig. 1 | The orthogonality of molecular orbitals and QI of 1Pb perovskite cluster. f, The corresponding transmission functions when lead R is attached to

different sites from 2 to 8.”

(4) The explanation of the sign dependence in quantum interference is a bit unclear. The text is too long and hard to understand. Authors may consider use one equation to explain it concisely (see *Acc. Chem. Res.* 2012, 45, 9, 1612-1621).

Response: Again, thank you very much for your valuable comments. We have removed these sentences from the previous version:

“As an example of this sign dependence, if the electrodes make contact with the left and right ends of the perovskite cluster in Fig. 1b, then QI is controlled by the signs of the HOMO and LUMO at each end of the cluster. The LUMO in Fig. 1b has a positive amplitude on the left (yellow) and a negative amplitude on the right (blue), hence the product of the left and right amplitudes (a_L) is negative. On the other hand, the HOMO has a positive amplitude on the left and a positive amplitude on the right, hence the product of the left and right amplitudes (a_H) is positive. As discussed in our previous papers,^{18,19} if the orbital products a_L and a_H have the opposite signs (as in Fig. 1b), then they will be constructive quantum interference (CQI) (high conductance) and when they have the same sign, they will be destructive quantum interference (DQI) (low conductance).”

To accommodate your suggestion, we referred to the literature you mentioned (*Acc. Chem. Res.* 2012, 45, 9, 1612-1621) and changed the explanation of quantum interference effects as follows:

“As mentioned in previous literature,¹⁸⁻²⁰ if the coupling between molecule and electrode is weak, the effect of quantum interference on transport properties could be predicted by examining the Green’s function $G(E_F)$ of the isolated molecule. The transmission amplitude of an electron with energy E_F from site i to j is proportional to $G_{i,j}(E_F) = \sum_{n=1}^N \frac{\phi_i^n \phi_j^n}{E_F - \varepsilon_n}$, where ϕ_i^n is the amplitude of n^{th} molecular orbital (MO) on site i and ε_n is the corresponding MO energy level. Taking only the HOMO and LUMO into consideration and assuming that E_F is located in the midgap of HOMO and

LUMO, this equation could be further written as $G_{i,j}(E_F) \approx \frac{1}{\Delta}(\phi_i^{HOMO} \phi_j^{HOMO} - \phi_i^{LUMO} \phi_j^{LUMO}) = \frac{1}{\Delta}(a_H - a_L)$, where Δ is half of the gap of HOMO and LUMO, $a_H = \phi_i^{HOMO} \phi_j^{HOMO}$, $a_L = \phi_i^{LUMO} \phi_j^{LUMO}$. Therefore, constructive quantum interference (CQI) corresponding to a large value of $|G_{i,j}(E_F)|$ is predicted if a_H and a_L have opposite signs, while destructive quantum interference (DQI), corresponding to a low value of $|G_{i,j}(E_F)|$, is predicted if a_H and a_L have the same sign. As an example of this sign dependence, if the electrodes make contact with the left and right ends of the perovskite cluster in Fig. 1b. The LUMO in Fig. 1b has a positive amplitude on the left (yellow) and a negative amplitude on the right (blue), hence the product (a_L) is negative. On the other hand, the HOMO has a positive amplitude on the left and a positive amplitude on the right, hence the product (a_H) is positive. Therefore, CQI is expected.

18. Lambert, C.J. & Liu, S.X. A magic ratio rule for beginners: a chemist's guide to quantum interference in molecules. *Chem.–Eur. J.* **24**, 4193-4201 (2018).

19. Zhao, X., Geskin, V. & Stadler, R. Destructive quantum interference in electron transport: A reconciliation of the molecular orbital and the atomic orbital perspective. *J. Chem. Phys.* **146**, 092308 (2017).

20. Yoshizawa, K. An orbital rule for electron transport in molecules. *Acc. Chem. Res.* **45**, 1612-1621 (2012).”

(5) What is the dash line in Fig. 2d and Fig. 3e? Does it indicate the slope of the jump?

I do not see any discussions about this in this current manuscript.

Response: Sorry for the misleading. The red dash lines in Fig. 2d and Fig. 3e are plotted as a guide to the eye, to indicate regions of increasing conductance. Since the latter regions are clear, we have removed the lines from the current version:

Fig. 2 | MCBJ measurements of MAPbBr₃ QDs. **a**, Schematic of MCBJ experimental principle in MAPbX₃ QDs. (MA= CH₃NH₃⁺, X=I⁻, Br⁻, Cl⁻, mixture of Br⁻ and Cl⁻.) The yellow arrows indicate the sliding process of the gold electrodes during the MCBJ measurements. **b**, Typical individual conductance-distance traces of pure solvent (green) and MAPbBr₃ QDs (blue) (the other conductance-distance traces are shown in Fig. S12). The jump plateaus are shown by red frame. **c**, All-data-points 2D conductance versus relative distance (Δz) histogram of MAPbBr₃ QDs (~3400 traces) and selected one conductance-distance trace. **d**, 2D relative conductance (G) versus relative displacement (Δz) histogram of the “jump curves” (~ 2400 traces) at the conductance-distance regime marked in Fig. 2c.

Fig. 3 | MCBJ measurements of MAPbBr₃, MAPbBr_{2.15}Cl_{0.85}, MAPbCl₃ and MAPbI₃ QDs. **a**, Typical individual conductance-distance traces of pure solvent, MAPbBr_{2.15}Cl_{0.85}, MAPbCl₃ and MAPbI₃. **b**, 1D Conductance histogram constructs without data selection for MAPbBr₃, MAPbBr_{2.15}Cl_{0.85}, MAPbCl₃ and MAPbI₃ QDs. The average diameters of MAPbBr₃ QDs are 6.34 nm (Fig. S3b) and 3.75 nm (Fig. S2), respectively. The conductance-distance traces are recorded ~2500 traces. **c**, The displacement distributions of three plateaus for MAPbBr₃ (up) and MAPbBr_{2.15}Cl_{0.85} (bottom). **d**, All-data-points 2D conductance versus relative distance (Δz) histogram for MAPbBr_{2.15}Cl_{0.85} and selected individual conductance-distance trace. **e**, 2D relative conductance (G) versus relative displacement (Δz) histogram of the “jump curves” (~1400 traces) for MAPbBr_{2.15}Cl_{0.85}. **f**, Raman spectra of Au-Br interaction on the gold substrates with SHINERS nanoparticles.”

Reviewer: 2

Comments:

In this manuscript, Zheng et al. reported a combined experimental and theoretical work on the quantum interference effect by investigating the conductance, and three distinct conductance features are observed from Br contained nanocrystals (NCs), while in Cl or I based samples, there are no such features. Overall, the experimental finding of the

conductance features in Br contained perovskite NCs is very interesting, however, based on my opinion, the proposed explanation or origin of these features is quite vague, therefore, I cannot recommend it for publication in its current form, the details are listed below:

(1) The binding detail of between gold and perovskite NC (or quantum dot QD) is questionable, clearly, n-octylamine and oleic acid are used to stabilize the NC, such that these organic ligands occupy the outer space of perovskite NCs, although the formation of Au-Br bonds are independently supported by Raman spectra and the conductance of oleic acid and octylamine are provided in SI, this Au-Br could be from the surface Br ions along with these organic ligands, or the migration of Br ions:

Response: Thank you for these valuable comments. However, even though the ligands occupy the outside of the QDs, they do not have strong and stable binding sites with gold electrodes and cannot be captured to form a single-molecule junction. This has been further confirmed by the control experiments of ligands and raw materials in SI (Fig. S10). The obvious molecular peaks can be observed in PbBr_2 , while no distinct conductance signal can be observed in other ligands and raw materials. Therefore, it is reasonable to conclude that the conductance plateaus obtained by our MCBJ measurements only originate from the strong Au-Br interaction and the organic ligands have no distinct impact on the charge transport process of the QDs.

In addition, it is impossible that the Au-Br bond confirmed by shell-isolated nanoparticle-enhanced Raman spectroscopy (SHINERS) originates from the surface Br ions along with these organic ligands, or the migration of Br ions. According to the MCBJ experimental results shown in Fig. 3b and 3c, the MAPbBr_3 and $\text{MAPbBr}_{2.15}\text{Cl}_{0.85}$ QDs present three clear and regular conductance plateaus and all the histograms are plotted without any data selection. The relative displacement distributions also show that the difference of adjacent statistical lengths matches well with the adjacent lattice distance of Br, confirming that it is the Br atoms occupied the corners of octahedral crystal cells provide the binding sites for Au-QD-Au junction. If the conductance signals come from the surface Br ions or the migration of Br ions, it is

impossible to obtain the regular conductance plateaus and even observe the obvious “jump conductance”. The displacement should also be randomly distributed.

Furthermore, some papers have proved that metal-halide interaction can form a very stable single-molecule junction (Tan, N., *et al. J. Am. Chem. Soc.* **138**, 679-687 (2016)). However, the CH_3NH_3^+ (MA^+) is located at the center of the regular octahedron, which is not easy to connect to the gold electrodes. The adjacent distance of the MA^+ is not in accordance with the displacement distributions. The electronegativity of the Pb^{2+} is low, and the Pb^{2+} is hidden within the Br networks that could not have reliable interaction with the gold electrodes. Therefore, both theoretical calculation and single-molecule conductance measurements suggests that the conductance plateaus of the single-QD junction originate from the Au-Br interaction sliding on the surface of the different lattice sites rather than the surface Br ions along with the ligands or the migration of Br ions.

To clarify this point, we have added these sentences in the manuscript:

“As for the other atoms, the MA^+ is located at the center of the regular octahedron, which is not easy to connect to the gold electrodes, and the adjacent distance of the MA^+ is not in accordance with the displacement distributions. The electronegativity of the Pb^{2+} is low, and the Pb^{2+} is hidden within the Br networks that could not have reliable interaction with the gold electrodes. Therefore, the gold electrodes interact with halogen, rather than other atoms or groups, to form stable Au-QD-Au junctions.”

(2) I also have concerns about the QD structures in the computation, as mentioned above, the information of organic ligands is completely missing, can these structures represent the ones in the experiments? Intuitively, I think the organic ligands have impacts on the interaction between gold and QDs, also the electronic structure of QDs.

Response: Thanks for your suggestion. In order to address this question, we carried out new calculations of 12Pb with ligands to prove the negligible influence of the ligands on the charge transport process of QDs. Five typical conformations were considered: a. 1 octylamine, b. 2 octylamines, c. 1 oleic acid, d. 2 oleic acids, e. 1 octylamine and 1

oleic acid. All of these five cases reveal that the ligands barely influence the transmission function, because of the weak interaction between the ligands and the perovskite quantum dot. As shown in Fig. S32, we place the ligand molecules close to the cluster and then relax these molecules, while freezing the perovskite cluster and the gold atoms. The transmission functions are presented on the right side in Fig. S32. For the oleic acid, the transmission function is nearly the same as the bare 12Pb cluster. For octylamine, a resonance appears at the position close to HOMO. However this will not influence the conductance since it is far from the Fermi energy. We also considered the case in which the ligand molecule bridges the gold electrode and the cluster to investigate the influence of ligand molecule on the coupling. The results are shown in Fig. S33a and b. In Fig. S33c, another connectivity is investigated, proving that the ligands have also no influence on the destructive quantum interference effects of the perovskite clusters. Our simulations demonstrate that the influence of the ligand is negligible over a large energy window in the HOMO-LUMO gap, due to the weak interaction between the ligand molecule and gold lead.

In order to address this point, we added Fig. S32 and S33 in **SI** and also these sentences in manuscript:

“The influence of ligand (oleic acid and octylamine) on the transmission functions were investigated by considering the ligand staying close to the cluster or bridging the gold lead and perovskite cluster (see Fig. S32 and S33 in SI). Our results reveal that the effect of the ligand is negligible due to the weak coupling between ligand molecules and cluster or gold lead.”

Fig. S32 | The influence of ligands on transmission functions. The conformations with ligands and the corresponding transmission functions are presented on the left side and right side respectively. For comparison, the transmission functions without ligand are also shown on the right side in blue color. **a**, 1 oleic acid **b**, 2 oleic acid **c**, 1 octylamine **d**, 2 octylamine **e**, 1 oleic acid and 1 octylamine.

Fig. S33 | The influence of ligands on transmission functions with bridge conformations and another connectivity. The situation of the organic ligands bridging gold lead and the cluster is investigated. The conformations and the corresponding transmission functions are presented on the left side and right side respectively. For comparison, the transmission functions without ligands are also shown on the right side in blue color. **a**, 1 oleic acid **b**, 1 octylamine **c**, 1 octylamine for another connectivity.”

(3) The authors provided statistical data on I, Cl, Br and Br/Cl samples, but the repeatability of the data on a specific sample, for example Br is not well supported, it would be interesting to see the experimental data on a few different pure Br based QD

samples;

Response: Thanks for these valuable comments. To ensure the repeatability of our experimental data, we carried out the break junction experiments and analyzed the data with the well-established protocol and high standards in the single-molecule break junction community that all results in the articles have been repeated at least three times, and all experimental data has been analyzed without any data selection. The conductance histograms and the displacement distributions in our article are widely used methods to describe the electrical properties and length information of single molecule junctions. (Venkataraman, L., *et al. Nature*, **558**, 415-419 (2018); Tao, N., *et al. Nat. Nanotechnol.* **13**, 316-321 (2018); van der Zant, H., *et al. Nat. Chem.* **10**, 1001-1007 (2018); Nichols, R., *et al. J. Am. Chem. Soc.* **140**, 710-718 (2018); Nuckolls, C., *et al. J. Am. Chem. Soc.*, **138**, 7791-7795 (2016)), and our single-molecule break junction investigations a been well recognized by the community. (Bai, J., *et al. Nat. Mater.* **18**, 364–369 (2019); Tan, Z., *et al. Nat. Commun.* **10**, 1748 (2019); Liu, J., *et al. Chem*, **5**, 390-401 (2019); Huang, X., *et al. Sci. Adv.* **5**, eaaw3072 (2019); Cai, S., *et al. Angew. Chem. Int. Ed.* **131**, 3869-3873 (2019)) Therefore, we believe our experimental data is convincing, highly repeatable and supports the conclusions of this work.

Furthermore, in order to confirm that the measurement is robust and highly reproducible, we also carried out the MCBJ measurements on a few pure Br-based QD samples with different sizes and diameters. We obtained the MAPbBr₃ perovskite QDs at the centrifugal speed of 5000 and 10000 rpm, respectively (as shown in Fig. S17). The experimental results show that perovskite quantum dots with different sizes display similar electrical properties, which proves that the conductance plateaus we measured originate from the perovskite crystal cell and the sizes and diameters of QDs do not affect the single-molecule experimental results. These results also give strong evidence that our experimental data is highly repeatable and well supported.

To give a more detailed demonstration, we have added the following sentences and the corresponding figures to the SI as section S3.9:

“**S3.9 MCBJ measurements of MAPbBr₃ QDs with centrifugal speeds**”

In order to prove the sizes and diameters of QDs have no clear impact on their electrical properties, we carry out the MCBJ measurements using the perovskite quantum dots obtained with the centrifugal speeds of 5000 rpm and 10000 rpm (as shown in Fig. S3). The MAPbBr₃ QDs centrifuged with different centrifugal speeds express similar conductance values located at $10^{-1.54}$, $10^{-2.80}$ and $10^{-4.32}$ for 5000 rpm, $10^{-1.43}$, $10^{-2.72}$ and $10^{-4.28}$ for 10000 rpm respectively, and the difference of adjacent statistical lengths matches well with the adjacent lattice distance of Br, which prove that the conductance plateaus we measured originate from the perovskite crystal cells rather than the entire perovskite QDs.

Fig. S17 | The MCBJ measurements of MAPbBr₃ QDs with different centrifugal speeds. **a.** 1D Conductance histogram constructs without data selection for MAPbBr₃ with the centrifugal speed of 5000 rpm. The conductance-distance traces are recorded ~2500 traces. **b.** All-data-points 2D conductance versus relative distance (Δz) histogram for MAPbBr₃ with the centrifugal speed of 5000 rpm. **c.** The displacement distributions of three plateaus for MAPbBr₃ with the centrifugal speed of 5000 rpm. **d.** 1D Conductance histogram constructs without data selection for MAPbBr₃ with the centrifugal speed of 10000 rpm. The conductance-distance traces are recorded ~2500 traces. **e.** All-data-points 2D conductance versus relative distance (Δz) histogram for MAPbBr₃ with the centrifugal speed of 10000 rpm. **f.** The displacement distributions of three plateaus for MAPbBr₃ with the centrifugal speed of 10000 rpm.”

We also added the following explanation in the manuscript:

“In addition, we also carry out the MCBJ measurements using the **MAPbBr₃** QDs with the average diameters of 6.34 nm and 3.75nm, which are obtained from the centrifugal speeds of 10000 rpm and 5000 rpm, respectively (as shown in Fig. S3 and Fig. S17). The experimental results show that the QDs with different diameters show similar conductance features, indicating that the conductance plateaus we measured originate from the perovskite crystal cells rather than the entire perovskite QDs.”

Furthermore, in order to further confirm the reproducibility of the experiments, we repeated the MCBJ measurements several times and the summaries of the experimental results for MAPbBr₃ and MAPbBr_{2.15}Cl_{0.85} QDs are shown in Table S1 and S2. From these experimental results, three or four clear conductance peaks can be observed with small standard deviations, suggesting that the experimental data is convincing and highly repeated.

We have added the following to the SI:

“**S3.10 The supplementary data of 1D conductance histograms of MAPbBr₃ QDs**

In order to confirm that the experimental data we measured has high repeatability and quality, we repeated the MCBJ measurements of MAPbBr₃ QDs several times at different bias voltages. The experimental data presents that all 1D conductance histograms display three or four clear conductance plateaus with small standard deviations, suggesting that the experimental data is highly reproducible and robust.

Fig. S18 | 1D conductance histograms of MAPbBr₃ QDs at different bias voltages.

Fig. S19 | 1D conductance histograms of MAPbBr_{2.15}Cl_{0.85} QDs at different bias voltages.

Table S1 | The summary of experimental results for MAPbBr₃ QDs

Conductance / log(G/G ₀)				
	P1	P2	P3	P4
Bias voltage / mV				
50mV	-1.57	-2.69	-4.07	
	-1.37	-2.46	-3.78	-5.23
	-1.37	-2.67	-4.04	-5.62
	-1.49	-2.64	-4.15	
AVE ¹	-1.45	-2.62	-4.01	-5.43
STD ²	0.08	0.09	0.14	0.20
100mV	-1.56	-2.75	-4.15	
	-1.56	-2.81	-4.2	
	-1.54	-2.73	-4.11	
	-1.5	-2.92	-4.17	
AVE	-1.54	-2.80	-4.16	
STD	0.02	0.07	0.03	
150mV	-1.64	-2.87	-4.24	-5.73
	-1.54	-2.8	-4.2	-5.67
	-1.44	-2.8	-4.16	-5.57
	-1.5	-2.78	-4.29	-5.6
AVE	-1.53	-2.81	-4.22	-5.64
STD	0.07	0.03	0.05	0.06
200mV	-1.67	-2.94	-4.36	
	-1.65	-2.9	-4.4	
	-1.58	-2.98	-4.49	
	-1.52	-2.78	-4.36	
AVE	-1.61	-2.90	-4.40	
STD	0.06	0.07	0.05	
250mV	-1.83	-3.05	-4.47	
	-1.7	-3.05	-4.43	
	-1.81	-3.14	-4.49	
	-1.84	-3.26	-4.63	
AVE	-1.80	-3.13	-4.51	
STD	0.06	0.09	0.08	

Table S2 | The summary of experimental results for MAPbBr_{2.15}Cl_{0.85} QDs

Conductance / log(G/G ₀)				
	P1	P2	P3	P4
Bias voltage / mV				
50mV	-1.52	-2.79	-4.35	
	-1.60	-3.01	-4.58	
	-1.41	-2.65	-4.01	-5.4

AVE	-1.51	-2.82	-4.31	
STD	0.08	0.15	0.23	
100mV	-1.51	-2.81	-4.23	
	-1.56	-2.93	-4.60	
	-1.56	-2.92	-4.33	-5.57
AVE	-1.54	-2.89	-4.39	
STD	0.02	0.05	0.16	
150mV	-1.61	-2.91	-4.34	
	-1.76	-3.09	-4.54	
AVE	-1.69	-3.00	-4.44	
STD	0.08	0.09	0.10	
200mV	-1.65	-2.91	-4.39	
	-1.77	-3.17	-4.70	
AVE	-1.71	-3.04	-4.55	
STD	0.06	0.13	0.16	
250mV	-1.64	-2.92	-4.38	
	-1.84	-3.20	-4.72	
AVE	-1.74	-3.06	-4.55	
STD	0.10	0.14	0.17	

1. AVE = The average of conductance values.

2. STD = The standard deviation of conductance values.”

(4) The authors show the theoretical results agrees well with the experimental, however, a comparison between the theoretical results also on Cl and I based QDs is needed to distinguish the Br QDs to the Cl and I based ones.

Response: Many thanks for this valuable comment. To address this point, we have carried out DFT calculations for the 12Pb cluster with Cl and I. Our results show that the three halide perovskite QDs (Cl, Br, I) possess similar charge transport properties (as shown in Fig. S34 and Fig. S35). If the Cl/I-based single-QD junctions could be stable and well-formed during the MCBJ measurements, the measured conductance evolutions of MAPbCl₃ and MAPbI₃ QDs would be similar to the MAPbBr₃ QDs. However, as mentioned in the manuscript, in a real experiment they are not expected to form stable junctions. As for the MAPbCl₃ QDs, according to the DFT calculation of Au-halogen binding energy in Table S3, the Cl-based QDs may not be able to be captured by the gold electrodes during the pulling process due to the weak Au-Cl bonding energy. In addition, due to the poorer stability of the MAPbI₃ perovskite QDs,

the crystal structure of the MAPbI₃ can be easily broken by the repeated pulling process of the gold electrodes thousands of times during the MCBJ measurements. (Yuan N., *et al. J. Mater. Chem. A* **3**, 5360-5367 (2015); Ghizi M., *et al. J. Phys. Chem. C* **121**, 27059-27070 (2017); Cahen D., *et al. MRS Communications* **5**, 623-629 (2015); Sit P., *et al. J. Phys. Chem. C* **119**, 22370-22378 (2015)).

To clarify this point, we added these sentences in the manuscript:

“We also carry out DFT calculations for MAPbCl₃ and MAPbI₃ QDs. Our results show that the three halide perovskite QDs possess similar charge transport features (see Fig. S26 and S27 in SI). However, as mentioned above, in a real experiment they are not expected to form junctions, because of the weaker Au-Cl bond and the poorer stability of crystal structure for MAPbI₃ QDs.²⁴⁻²⁶”

24. Faghihnasiri, M., Izadifard, M. & Ghazi, M.E. DFT Study of mechanical properties and stability of cubic methylammonium lead halide perovskites (CH₃NH₃PbX₃, X = I, Br, Cl). *J. Phys. Chem. C* **121**, 27059-27070 (2017).

25. Rakita, Y., Cohen, S.R., Kedem, N.K., Hodes, G. & Cahen, D. Mechanical properties of APbX₃ (A = Cs or CH₃NH₃; X = I or Br) perovskite single crystals. *MRS Commun.* **5**, 623-629 (2015).

26. Dong, X., *et al.* Improvement of the humidity stability of organic–inorganic perovskite solar cells using ultrathin Al₂O₃ layers prepared by atomic layer deposition. *J. Mater. Chem. A* **3**, 5360-5367 (2015).”

In addition, we add Fig. S34-S35 and the following sentences in Computational methods part to SI (S4.2.1):

“After relaxation, the optimized separation between contact halogen atoms (Cl, Br, I) and apex gold atom were found to be 2.66 Å, 2.76 Å and 2.88 Å respectively.”

Fig. S34 | DFT calculations of 12Pb MAPbCl₃ cluster. **a**, Conformations for 12Pb MAPbCl₃ cluster embedded in two gold electrodes with 4 different connectivity L-R1, L-R2, L-R3 and L-R4. **b**, The corresponding transmission functions and room temperature conductances with the $E_F = -1$ eV relative to that estimated by DFT which is indicated by the black dashed line in the left panel.

Fig. S35 | DFT calculations of 12Pb MAPbI₃ cluster. a, Conformations for 12Pb MAPbI₃ cluster embedded in two gold electrodes with 4 different connectivity L-R1, L-R2, L-R3 and L-R4. **b,** The corresponding transmission functions and room temperature conductances with the $E_F = -1$ eV relative to that estimated by DFT which is indicated by the black dashed line in the left panel.”

Reviewer: 3

Comments:

This manuscript has demonstrated experimental and theoretical investigation of room temperature QI effects in the electron transport through single perovskite QD junctions. Three distinct conductance features are observed from the conductance measurements

of perovskite QDs with Br, while the QDs with I and Cl show no significant features. The multiple conductance features are derived from the sliding of gold electrodes between the adjacent Br atoms in different unit cells. A distinct conductance jump at the end of individual conductance traces, which is claimed as room-temperature QI effects. Basically this is interesting work and could possibly published in Nature Communications only after the following points are considered.

(1) The distinct conductance jump at the end of individual conductance traces is very interesting and owing to the junction switch from "L-R3" to "L-R4". However, this is ideal case only for three lattices. From the TEM images of Fig. S2 and S3, the QDs are not uniform enough. To support the authors' conclusion, more cases with different numbers of lattices should be included. Then the conductance jumps may appear after two or four (and so on) conductance plateaus.

Response: Thanks for this constructive comment. First of all, from the TEM image in the SI, we agree with your statement that the QDs are not uniform. In other words, these QDs have various sizes and diameters. In fact, it is difficult to guarantee the QDs are uniform and all have similar diameters in the synthesis process even though the synthetic experiments were carried out in the glovebox and controlled the water and oxygen in the environment strictly. (Dong Y. , *et al. ACS Nano* **9**, 4533-4542 (2015); Andrey R., *et al. Adv. Sci.* **2**, 1500194, (2015); Zhong H., *et al. Chem. Mater.* **298**, 3793-3799 (2017)) However, non-uniform QDs actually would present similar conductance features. For example, from the HRTEM images of the MAPbBr₃ and MAPbBr_{2.15}Cl_{0.85} QDs, although these two kinds of QDs display a significant difference in diameter distributions (3.75 ± 1.39 nm and 6.32 ± 1.63 nm, respectively), as shown in Fig. 3b, there is little discernible difference in their charge transport properties (three clear conductance plateaus and almost same conductance values), indicating that the diameters and sizes of the QDs have no distinct impact on their electrical properties. This can be attributed to the regular and cubic crystal structure of the perovskite QDs. According to our experimental analysis and theoretical explanation, the three conductance plateaus originate from the perovskite crystal cells through Au-Br

interaction rather than the entire QDs, so the sizes and diameters of QDs are independent on the experimental results. It is because the average displacement differences of the three plateaus (~ 0.5 nm) match well with the adjacent lattice distances of Br (~ 0.5 nm) (Pérez-Prieto, J., *et al. J. Am. Chem. Soc.* **136**, 850-853 (2014); Zhao Y., *et al. Chem. Commun.* **51**, 7820-7823, (2015)) rather than the diameters of the QDs (3.75 nm or 6.32 nm).

To further verify the above conclusions, we carried out the MCBJ measurements of MAPbBr₃ QDs obtained from the centrifugal speeds of 5000 and 10000 rpm, respectively (as shown in Fig. S3). The HRTEM images suggest that the QDs obtained from the centrifugal speeds of 10000 rpm display the similar average diameters (6.34 nm) to the centrifugal speeds of 5000 rpm (6.35 nm), while the former has the smaller standard deviation (1.35 nm) than the latter (2.27 nm), further proving that although the QDs are not non-uniform, the diameters and distributions of QDs do not affect the MCBJ experimental results.

To give a more detailed demonstration, we add these sentences and the corresponding figures to the SI:

Fig. S3 | The image of transmission electron microscopy for MAPbBr₃ perovskite QDs with different centrifugal speeds. a, The HRTEM image for MAPbBr₃ perovskite QDs with the centrifugal speed of 5000 rpm. b, The HRTEM image for MAPbBr₃ perovskite QDs with the centrifugal speed of 10000 rpm.

S3.9 MCBJ measurements of MAPbBr₃ QDs with centrifugal speeds

In order to prove the sizes and diameters of QDs have no impact on their electrical properties, we carry out the MCBJ measurements using the perovskite quantum dots obtained with the centrifugal speeds of 5000 rpm and 10000 rpm (as shown in Fig. S3). The MAPbBr₃ QDs centrifuged with different centrifugal speeds express similar conductance values located at $10^{-1.54}$, $10^{-2.80}$ and $10^{-4.32}$ for 5000 rpm, $10^{-1.43}$, $10^{-2.72}$ and $10^{-4.28}$ for 10000 rpm, respectively, and the difference of adjacent statistical lengths matches well with the adjacent lattice distance of Br, which prove that the conductance plateaus we measured originate from the perovskite crystal cells rather than the entire perovskite QDs.

Fig. S17 | The MCBJ measurements of MAPbBr₃ QDs with different centrifugal speeds. **a.** 1D Conductance histogram constructs without data selection for MAPbBr₃ with the centrifugal speed of 5000 rpm. The conductance-distance traces are recorded ~2500 traces. **b.** All-data-points 2D conductance versus relative distance (Δz) histogram for MAPbBr₃ with the centrifugal speed of 5000 rpm. **c.** The displacement distributions of three plateaus for MAPbBr₃ with the centrifugal speed of 5000 rpm. **d.** 1D Conductance histogram constructs without data selection for MAPbBr₃ with the centrifugal speed of 10000 rpm. The conductance-distance traces are recorded ~2500 traces. **e.** All-data-points 2D conductance versus relative distance (Δz) histogram for

MAPbBr₃ with the centrifugal speed of 10000 rpm. **f.** The displacement distributions of three plateaus for MAPbBr₃ with the centrifugal speed of 10000 rpm.”

In addition, we also added the following explanation in the manuscript:

“In addition, we also carry out the MCBJ measurements using the **MAPbBr₃** QDs with the average diameters of 6.34 nm and 3.75 nm, which are obtained from the centrifugal speeds of 10000 rpm and 5000 rpm, respectively (as shown in Fig. S3 and Fig. S17). The experimental results show that the QDs with different diameters display similar conductance features, indicating that the conductance plateaus we measured originate from the perovskite crystal cells rather than the entire perovskite QDs.”

On the other hand, we agree with your statement that “more cases with different numbers of lattices should be included to support our conclusion”. Indeed, it is possible that the gold electrode could go along all directions and the conductance jumps may appear after two or four (and so on) conductance plateaus. Actually, from the theoretical results of Fig. S26 and Fig. S27 in **SI**, we have shown that the conductance jumps can appear after two or four conductance plateaus for 2x2x2 8Pb and 2x2x4 16Pb clusters in the previous version. To include more cases, we also calculated the transmission functions of 3x3x2 18Pb MAPbBr₃, 2x2x3(2) 10Pb MAPbBr₃ which are a bit irregular (obtained by removing two Pb units based on the 2x2x3 12Pb MAPbBr₃ and then relax it) and 2x2x3 12Pb MAPbBr₃ with the different binding sites of the left electrodes. More Pb units would be too computationally expensive since 18Pb MAPbBr₃ has 405 atoms already. Our theoretical results of the Br-Br distances and conductances were summarized in Table S4 (shown below). It is demonstrated that the conductances of clusters with different numbers of lattices display similar features (the conductance first decays exponentially and then jumps at end). However, the attenuation factors β of these different cases vary significantly (from 0.54 / Å to 1.2 / Å). In addition, when the distances between two gold electrodes are too great (such as cases in Fig. S29 and Fig. S38-S39), the conductances of MAPbBr₃ are below the detection limit of our instrument ($10^{-6} G/G_0$). The theoretical distances of these cases (> 15 Å) are also much larger than the experimental displacement distributions (maximum 12 Å). Therefore, the

theoretical cases with too low conductances and too long distances can be excluded from the experimental results, and our theoretical models with 18Pb clusters are sufficient to reflect the vast majority of possible experimental cases.

To further provide the possible theoretical models of the perovskite QD junctions, we added Table S4 and Fig. S36-S39 in SI and also replaced “This increase at the most distant electrode separation is also found in 8Pb and 16Pb **MAPbBr₃** clusters (Fig. S28 and Fig. S29)” in the manuscript by “This increase at the most distant electrode separation is also found in 8Pb, 10Pb (obtained by removing two Pb units based on the 2x2x3 12Pb **MAPbBr₃**), 16Pb and 18Pb **MAPbBr₃** clusters (Fig. S26-S31 and S36-S37).”

Fig. 36 | DFT calculations of 3x3x2 18Pb MAPbBr₃ cluster. a, Conformations for

18Pb MAPbBr₃ cluster embedded in two gold electrodes with 4 different connectivity L-R1, L-R2, L-R3 and L-R4. **b**, The corresponding transmission functions and room temperature conductances with the $E_F = -1$ eV relative to that estimated by DFT which is indicated by the black dashed line in the left panel.

Fig. 37 | DFT calculations of 2x2x3 10Pb MAPbBr₃ cluster. **a**, Conformations for 10Pb MAPbBr₃ cluster embedded in two gold electrodes with 4 different connectivity L-R1, L-R2, L-R3 and L-R4. **b**, The corresponding transmission functions and room temperature conductances with the $E_F = -1$ eV relative to that estimated by DFT which is indicated by the black dashed line in the left panel.

Table S4 The summary of theoretical results

Br-Br distance (Å)	R1	R2	R3	R4	R5	β (Å ⁻¹)
Conductance (log(G/G ₀))						
8Pb	5.09	9.85	12.57			

(Fig. S26)	-2.61	-4.46	-2.69			
12Pb (Fig. 4c)	4.77	9.55	15.40	18.47		0.54
16Pb (Fig. S27)	4.71	9.41	15.43	20.06	24.21	0.54
16Pb-cross (Fig. S28)	5.33	11.43	16.13	20.7		0.72
16Pb-cross (Fig. S29)	7.94	12.71	17.64	21.64		1.2
18Pb-diagonal (Fig. S31)	4.98	10.89	18.2	20.84	20.63	0.87
18Pb (Fig. S36)	4.71	9.38	15.39	18.28		0.55
10Pb (Fig. S37)	5.19	9.91	14.94	18.31		0.98
12Pb-L ^a (Fig. S38)	6.76	11.00	16.00	18.78		0.42
12Pb-L ^b (Fig. S39)	10.53	13.7	17.98	19.47		0.88

»

Referring to the above theoretical models, in order to analyze experimentally the possible binding sites and their relative proportions, we use the spectral clustering algorithm to give comprehensive and detailed classifications of the individual conductance-distance traces. The original conductance-distance traces can mainly be divided into five categories. As for MAPbBr₃ QDs: 15.5% of traces with three successive conductance plateaus, 35.0% of traces with the highest conductance (HC) and middle conductance (MC), 31.3% of traces with the HC and the lowest conductance (LC), 9.9% of traces with MC and LC, and 8.3% of traces with only MC. As for the MAPbBr_{2.15}Cl_{0.85} QDs: 16.2% of traces with three successive conductance plateaus, 44.0% of traces with HC and MC, 21.3% of traces with the HC and LC, 8.7% of traces with MC and LC, and 9.8% of traces with only MC. When the experimental environment and instrument condition are excellent, we indeed can even observe four distinct conductance plateaus with conductance “jump” at the end of the conductance traces (as shown in Fig. S22). Even though the conductance jumps can appear after two or four conductance plateaus, these conductance plateaus display similar conductance

features, ie the similar conductance values and displacement distributions, further confirming that the gold electrodes are more likely to slide along the ‘horizontal’ direction rather than the ‘diagonal’ direction.

Furthermore, comparing these theoretical conductances and Br-Br distances with our classified experimental results, it is clear that only the case that the gold electrodes first connect to the closest Br atoms and slide along the ‘horizontal’ direction can well support our experimental results (such as Fig. 4c, S27 and S36).

To clarify this point, we have added these sentences in the manuscript and **SI**:

“In order to further analyze the possible binding sites of the gold electrodes during the pulling process, we use the spectral clustering algorithm to give comprehensive and detailed classifications of the individual conductance-distance traces. The original conductance-distance traces can mainly be divided into five categories (as shown in Fig. S20 and S21). The results show that although the three-step plateaus do not always appear simultaneously, the classified conductance plateaus display similar conductance features, ie the similar conductance values and displacement distributions, further confirming that the gold electrodes are more likely to slide along the ‘horizontal’ direction rather than the ‘diagonal’ direction.”

“S3.11 The classification results of MAPbBr₃ and MAPbBr_{2.15}Cl_{0.85} QDs using unsupervised clustering algorithms

The spectral clustering algorithm is a state of art clustering technique which provides a partition of data and assigns similar data traces into clusters. Here we firstly review the spectral clustering algorithm according to the Ng *et al.*^[1] on the 1D conductance histograms.

Given histogram data $H = \{h_1, h_2, \dots, h_M\}$ in R^N (dividing the conductance axis to discrete N bins) that we want to cluster into K clusters:

1. Form the affinity matrix $A \in R^{M \times M}$ defined by $A_{ij} = C_{ij} + 1$ if $i \neq j$, and $A_{ii} = 0$.
2. Define D to be the diagonal matrix whose (i, j) -element is the sum of A 's i -th

row, and construct the matrix $L=D^{-1/2}AD^{-1/2} - I$.

3. Find x_1, x_2, \dots, x_K , the K largest eigenvectors of L , and form the matrix $X = [x_1, x_2, \dots, x_K]$ belong to $R^{M \times K}$ by stacking the eigenvectors in columns.
4. Treating each row of X as a point in R^K , cluster them into K clusters via K-means++.
5. Finally, assign the original points h_i to cluster j if and only if row i of the matrix X was assigned to cluster j .

Here we construct the affinity matrix A specified different from the usual one (the Gaussian (aka RBF) kernel), the other steps are almost the same as described in Ng *et al*^[1]. Here we define the C_{ij} as the cross-correlation between histogram h_i and h_j as follow:

$$C_{ij} = \frac{\langle [h_i - \langle h_i \rangle][h_j - \langle h_j \rangle] \rangle}{\sqrt{\langle [h_i - \langle h_i \rangle]^2 \rangle \langle [h_j - \langle h_j \rangle]^2 \rangle}} \quad (1)$$

where $\langle h_i \rangle$ represents the average value of histogram h_i , the values of C range from $[-1, 1]$, so we add one to make the elements of affinity matrix A nonnegative to meet the spectral clustering requirements. h_i is the conductance histogram for the i -th individual trace, M is the number of conductance traces, N is the number of the histogram bins.

Fig. S20 | The classification results of MAPbBr₃ QDs using spectral clustering algorithms. a, 15.5% of traces with three successive conductance plateaus. b, 35.0% of traces with the highest conductance (HC) and middle conductance (MC). c, 31.3% of traces with the HC and the lowest conductance (LC). d, 9.9% of traces with MC and LC. e, 8.3% of traces with only MC.

Fig. S21 | The classification results of MAPbBr_{2.15}Cl_{0.85} QDs using spectral clustering algorithms. a, 16.2% of traces with three successive conductance plateaus. b, 44.0% of traces with the highest conductance (HC) and middle conductance (MC). c, 21.3% of traces with the HC and the lowest conductance (LC). d, 8.7% of traces with MC and LC. e, 9.8% of traces with only MC.

Fig. S22 | The classification results of MAPbBr₃ QDs with four conductance plateaus.

plateaus using spectral clustering algorithms. **a**, 1D Conductance histogram constructs without data selection for MAPbBr₃ QDs. **b**, All-data-points 2D conductance versus relative distance (Δz) histogram for MAPbBr₃ QDs. **c**, The displacement distributions of three plateaus for MAPbBr₃ QDs. **d**, 2D relative conductance (G) versus relative displacement (Δz) histogram of the “jump curves” for MAPbBr₃ QDs.”

(2) Three exposed halogen atoms on the corner lattice had the opportunity to interact with the gold electrodes. One of them was labelled as "L" in this work. The authors only assumed and calculated one ideal case. How about the other two cases for the "L" atom in two orthogonal directions? The conclusion from just one special case is not convinced enough since the authors cannot precisely control the junction between the QD and the Au electrodes.

Response: Thanks for this nice suggestion. In order to simplify the theoretical models and reduce the computational burden, we fixed the left gold electrode and changed the sites of the right gold electrode to simulate the sliding process in the real experiments. In fact, from the theoretical results of Fig. S27-S29 in the previous version, we have shown three different fixed sites of the left gold electrode for 2x2x4 16Pb clusters and the results have been explained in the manuscript.

“Other possible connectivities for 16Pb MAPbBr₃ cluster are also explored (Fig. S27 and Fig. S28), we find that this jump behaviour is generic although different β factors are observed ($0.72/\text{\AA}$ and $1.2/\text{\AA}$ separately), and the latter connectivity is less likely to appear in the experiments due to the higher energy barrier.”

However, we only consider the cases that the two gold electrodes are connected to the two closest Br atoms at the beginning of this pulling process. This is because, in real experiments, after the rupture of the gold wire, the initial gap width is known to be a snap-back distance of about 5 Å. Since this corresponds to the distance between two closest Br atoms (around 5 Å), these two Br atoms are most likely to be connected to the gold atoms at the beginning in this pulling process.

Nevertheless, for completeness, two new left binding sites (L^a and L^b) were considered

in the new calculation. The gold electrodes in these cases are not connected to the two closest Br atoms at the beginning of the pulling process. As shown in Fig. S38 and Fig. S39, we find the conductance evolution follows the same trend as the case of the left contact Br atom (L), ie it first decays exponentially and then jumps at the end. However, the magnitude of conductances is much smaller in these two cases due to the higher barrier caused by the larger Br-Br distance. In addition, the conductances of the cases L^a -R1 and L^b -R1 are also much smaller than our experimental results. Therefore, it is reasonable to conclude that the gold electrodes are more likely to connect to the two closest Br atoms at the beginning of the junction formation.

To clarify this point, we added Fig. S38 and S39 in **SI**, and these sentences in manuscript:

“Two new left binding sites (L^a and L^b) were also considered in our calculations. As shown in Fig. S38 and Fig. S39, we find the conductance evolution follows the same trend as the binding sites of L, ie it first decays exponentially and then jumps at end. However, the magnitude is much smaller in these two cases due to the higher barrier caused by the larger Br-Br distance.”

Fig. 38 | DFT calculations of 2x2x3 12Pb MAPbBr₃ cluster with a new left contact Br atom 'L^a'. a, Conformations for 12Pb MAPbBr₃ cluster embedded in two gold electrodes with 4 different connectivity L^a-R1, L^a-R2, L^a-R3 and L^a-R4. b, The corresponding transmission functions and room temperature conductances with the $E_F = -1$ eV relative to that estimated by DFT which is indicated by the black dashed line in the left panel.

Fig. 39 | DFT calculations of 2x2x3 12Pb MAPbBr₃ cluster with the other new left contact Br atom ‘L^b’. a, Conformations for 12Pb MAPbBr₃ cluster embedded in two gold electrodes with 4 different connectivity L^b-R1, L^b-R2, L^b-R3 and L^b-R4. b, The corresponding transmission functions and room temperature conductances with the E_F = -1 eV relative to that estimated by DFT which is indicated by the black dashed line in the left panel.”

(3) The motivation for using MAPbBr_{3-x}Cl_x QDs should be stated. And why not MAPbBr_{3-x}I_x QDs? Moreover, the exact values of 3-x and x should be given in the formula.

Response: Thank you for your comments. Actually, due to the higher stability and the more regular crystal structure than the MAPbBr_{3-x}I_x QDs (Yuan N., *et al. J. Mater. Chem. A* **3**, 5360-5367 (2015); Ghizi M., *et al. J. Phys. Chem. C* **121**, 27059-27070 (2017); Cahen D., *et al. MRS Communications* **5**, 623-629 (2015); Sit P., *et al. J. Phys. Chem.*

C 119, 22370-22378 (2015)), the MAPbBr_{3-x}Cl_x QDs are the first choice to investigate the quantum interference effects and further confirm the binding sites through Au-Br interaction during the MCBJ measurements.

In addition, in order to provide further evidence that the Au-I interaction cannot form the stable single-QD junction, we carried out the MCBJ measurements with MAPbBr_{3-x}I_x QDs and added the following sentences in the SI as section S3.8:

“S3.8 MCBJ measurements of MAPbBr_{2.15}I_{0.85} QDs

In order to provide further evidence that the Au-I interaction cannot form the stable single-QD junction, we have also measured the electrical properties of MAPbBr_{2.15}I_{0.85} perovskite QDs. The experimental results show that neither the peak of gold-gold atomic junction nor the conductance plateaus of the single-QD junction can be observed in MAPbBr_{2.15}I_{0.85} QDs, which is similar to the charge transport properties of MAPbBr₃ QDs. Therefore, it can be demonstrated that the Au-I interaction cannot form stable single-QD junctions due to its stronger bond energy and poorer stability of the crystal structure.

Fig. S16 | The MCBJ measurement of MAPbBr_{2.15}I_{0.85} QDs. a, 1D conductance histograms for perovskite QDs MAPbBr_{2.15}I_{0.85}. b, The 2D conductance versus relative distance (Δz) histogram for perovskite QDs MAPbBr_{2.15}I_{0.85}.”

And we also revised this sentence in the manuscript to:

“To reveal the binding geometries of the single QD junctions, we carry out the single-QD conductance measurements of MAPbBr_{2.15}Cl_{0.85}, MAPbBr_{2.15}I_{0.85}, MAPbCl₃ and MAPbI₃ QDs.”

and

“.....We also construct the conductance histograms for **MAPbCl₃**, **MAPbI₃** and **MAPbBr_{2.15}I_{0.85}** from ~2500 individual traces (Fig. 3b and Fig. S16), and no conductance peak are observed, while the peak of the gold-gold atomic junction at G_0 for **MAPbI₃** and **MAPbBr_{2.15}I_{0.85}** becomes less clear than others.”

and

“.....In contrast, the strong Au-I bond may break the crystal structure of **MAPbBr_{2.15}I_{0.85}** and **MAPbI₃** with the pulling process of the electrodes.”

On the other hand, the exact values of 3-x and x have been given in our previous version in SI:

“... The above results confirm the mixed halide in **MAPbBr_{3-x}Cl_x** QDs and the approximate composition of **MAPbBr_{3-x}Cl_x** QDs is roughly **MAPbBr_{2.15}Cl_{0.85}**.”

In order to make this point clearer, we changed all the **MAPbBr_{3-x}Cl_x** and **MAPbBr_{3-x}I_x** to **MAPbBr_{2.15}Cl_{0.85}** and **MAPbBr_{2.15}I_{0.85}** in the present version.

(4) The authors claimed the junction is based on Au-Br interactions. Therefore, why no conductance plateaus can be observed for PbBr₂? I doubt the control experiment was not reasonably carried out.

Response: Thank you for this question. In the previous version, in order to maintain the consistency of the experimental conditions, we used a similar solvent, TMB, to carry out both the control measurements and the perovskite-QDs conductance measurements. However, this leads to the poor solubility of perovskite raw materials in TMB solvent. Therefore, we replace the solvent with γ -butyrolactone (GBL) to repeat all the control experiments and also use the ultrasonic concussion for at least 30 minutes to guarantee the solubility. (O. M. Bakr, *et al. Chem. Commun.* **51**, 17658-17661 (2015); H. L. Clever, *et al. J. Phys. Chem. Ref. Data.* **9**, 751-784. (1980)) The experimental results show that two clear molecular peaks can be observed in PbBr₂, while no conductance signal can be observed in other ligands and raw materials. The PbBr₂ would change to (PbBr₆)⁴⁻ and (Pb₂Br₉)⁵⁻ in the solvent of γ -butyrolactone,

corresponding to the two conductance peaks in Fig. S9g. (Barta, Č., *et al. J. Phys. B* **20**, 803-807 (1970); Berka, L., *et al. Anal. Chem.* **44**, 2192-2195 (1972); Gionis, V., *et al. J. Mater. Chem.* **8**, 2259-2262 (1998)) Therefore, it can be demonstrated that Au-Br binding is stable enough to form stable molecular junctions, while other atoms are relatively difficult to form stable molecular junctions.

To further clarify this point, we have added these sentences in the SI:

“To exclude the effects from the ligands and synthetic raw materials (oleic acid, octylamine, PbCl₂, PbBr₂, MACl and MABr) during the MCBJ measurement of perovskite QDs, the control experiments of all the ligands are carried out in the solvent, γ -butyrolactone (GBL), and also use the ultrasonic concussion for at least 30 minutes to guarantee the solubility.^{22,23} The 1D conductance histogram and 2D conductance-distance histogram are shown in Fig. S10. The obvious conductance plateau can be observed in PbBr₂, while no clear conductance signal can be observed in other ligands and raw materials, suggesting that the conductance signal only derives from Au-Br interaction. The PbBr₂ would change to (PbBr₆)⁴⁻ and (Pb₂Br₉)⁵⁻ in the solvent of GBL, corresponding to the two conductance peaks in Fig. S9g.²⁴⁻²⁶

21. Saidaminov, M.I., Abdelhady, A.L., Maculan, G. & Bakr, O.M. Retrograde solubility of formamidinium and methylammonium lead halide perovskites enabling rapid single crystal growth. *Chem. Commun.* **51**, 17658-17661 (2015).

22. Clever, H.L. & Johnston, F.J. The solubility of some sparingly soluble lead salts: an evaluation of the solubility in water and aqueous electrolyte solution. *J. Phys. Chem. Ref. Data* **9**, 751-784 (1980).

23. Bohun, A., Dolejší, J. & Barta, Č. The absorption and luminescence of (PbCl₆)⁴⁻ and (PbBr₆)⁴⁻ complexes. *Czechoslovak Journal of Physics B* **20**, 803-807 (1970).

24. Nergararian, A. & Berka, L. Anomalous absorption bands in ultraviolet spectra of halide solutions. *Anal. Chem.* **44**, 2192-2195 (1972).

25. Mousdis, G.A., Gionis, V., Papavassiliou, G.C., Raptopoulou, C. & Terzis, A. Preparation, structure and optical properties of [CH₃SC(=NH₂)NH₂]₃PbI₅, [CH₃SC(=NH₂)NH₂]₄Pb₂Br₈ and [CH₃SC(=NH₂)NH₂]₃PbCl₅·CH₃SC(=NH₂)NH₂Cl. *J. Mater.*

Fig. S10 | The MCBJ measurements of ligands and synthetic raw materials. a, 1D conductance histograms for oleic acid. **b,** The 2D conductance versus relative distance (Δz) histogram for oleic acid. **c,** 1D conductance histograms for octylamine. **d,** The 2D conductance versus relative distance (Δz) histogram for octylamine. **e,** 1D conductance histograms for PbCl_2 . **f,** The 2D conductance versus relative distance (Δz) histogram for PbCl_2 . **g,** 1D conductance histograms for PbBr_2 . **h,** The 2D conductance versus relative distance (Δz) histogram for PbBr_2 . **i,** 1D conductance histograms for MACl. **j,** The 2D conductance versus relative distance (Δz) histogram for MACl. **k,** 1D conductance histograms for MABr. **l,** The 2D conductance versus relative distance (Δz) histogram for MABr.”

And we also revise the sentences in the manuscript to:

“To reveal the source of the conductance plateaus, we also carry out the MCBJ

measurements of all ligands and ingredients used in the synthesis of the QDs in the solvent γ -butyrolactone, including oleic acid, octylamine, PbBr₂, PbCl₂, MAcl and MABr. The obvious conductance plateau can be observed in PbBr₂, while no clear conductance signal can be observed in other ligands and raw materials, suggesting that the conductance signal may come from Au-Br interaction and the other ligands cannot form the single-QD junction. (Fig. S10).”

(5) I do not believe Au-I bond is strong enough to break the stable crystal structure of MAPbI₃ with the pulling process of the electrodes. This explanation needs further support.

Response: Thank you for this comment. In our opinion, the crystal structure of the MAPbI₃ can be easily broken by the repeated pulling process of the gold electrodes thousands of times during the MCBJ measurements, which can be fully supported by the well-known term - “Goldschmidt tolerance factor (t)”: (Reaney I., *et al. J. Appl. Phys.* **33**, 3984-3990 (1994); Fu Y., *et al. Nat. Rev. Mater.* **4**, 169–188 (2016); Li Z., *et al. Chem. Mater.*, **28**, 284–292 (2016))

$$t = (R_A + R_X) / \sqrt{2}(R_B + R_X)$$

where R_A , R_B and R_o are the radii of the A- and B- site ions and the X-ion, respectively. The value of tolerance factor between 0.8 and 1 is favorable for cubic ($0.9 < t < 1$) or distorted ($0.8 < t < 0.9$) perovskite structures, which provides an empirical rule for predicting the stability and lattice distortion of perovskite structures. In this paper, A-site ion represents MA⁺. B-site ion represents Pb²⁺. X-site ion represents the halide anion (Br⁻, Cl⁻ or I⁻). The tolerance factors of the halide perovskite structures in this paper are calculated to be 0.927 for MAPbBr₃, 0.938 for MAPbCl₃ and 0.911 for MAPbI₃ respectively, which proves that the MAPbI₃ perovskite has the most distorted crystal structure and the poorest stability. In addition, various papers also confirm that the MAPbI₃ perovskite has poorer stability of crystal structure and higher sensibility to the water and oxygen. (Yuan N., *et al. J. Mater. Chem. A* **3**, 5360-5367 (2015); McGehee M., *et al. Chem. Rev.* **119**, 3418-3451 (2019); Cahen D., *et al. MRS Communications* **5**, 623-629 (2015); Haque S., *et al. Energy Environ. Sci.* **9**, 1655-1660

(2016); Zhou H., *et al. Energy Environ. Sci.* **10**, 2284-2311 (2017); Sit P., *et al. J. Phys. Chem. C* **119**, 22370-22378 (2015); Melot B., *et al. Chem. Commun.* **50**, 15819-15822 (2014)) Therefore, the MAPbI₃ crystal structure is more likely to be broken by the pulling process of the gold electrodes thousands of times during the MCBJ measurements, which provides the evidence that the 1D and 2D conductance histograms of the MAPbI₃ display neither conductance plateaus nor gold peaks. As for the MAPbCl₃, although it has the most stable crystal structure, the poor strength of the Au-Cl bond makes it difficult to be captured by the gold electrodes during the pulling processes. Therefore, the gold peaks of the MAPbCl₃ can be observed in 1D and 2D conductance histograms but no conductance plateaus.

To further clarify this point, we revised the sentence in the manuscript to:

“In contrast, the strong Au-I bond may break the crystal structure of **MAPbBr_{2.15}I_{0.85}** and **MAPbI₃** with the pulling process of the electrodes **due to the poorer stability of crystal structure.**²⁴⁻²⁶

24. Faghihnasiri, M., Izadifard, M. & Ghazi, M.E. DFT Study of mechanical properties and stability of cubic methylammonium lead halide perovskites (CH₃NH₃PbX₃, X = I, Br, Cl). *J. Phys. Chem. C* **121**, 27059-27070 (2017).

25. Rakita, Y., Cohen, S.R., Kedem, N.K., Hodes, G. & Cahen, D. Mechanical properties of APbX₃ (A = Cs or CH₃NH₃; X = I or Br) perovskite single crystals. *MRS Commun.* **5**, 623-629 (2015).

26. Dong, X., *et al.* Improvement of the humidity stability of organic–inorganic perovskite solar cells using ultrathin Al₂O₃ layers prepared by atomic layer deposition. *J. Mater. Chem. A* **3**, 5360-5367 (2015).”

REVIEWERS' COMMENTS:

Reviewer #1 (Remarks to the Author):

The authors carried out more experiments on different quantum dots to support their conclusions that the experimental results do not depend on the morphology of QDs. The authors also performed extra data analysis using spectral clustering algorithms to classify their individual traces. I found these new results are convincing. I would recommend for publication. I also suggest the authors to share their algorithms on some public resources since the algorithms are important to the manuscript.

Reviewer #2 (Remarks to the Author):

In this revised manuscript, all my concerns have been addressed, I would like to recommend it for publication in its current form.

Reviewer #3 (Remarks to the Author):

The comments from the reviewers have been well considered and the manuscript is now valuable to be published.

Point-to-point reply to reviewers' comments

Reviewer: 1

Comments:

The authors carried out more experiments on different quantum dots to support their conclusions that the experimental results do not depend on the morphology of QDs. The authors also performed extra data analysis using spectral clustering algorithms to classify their individual traces. I found these new results are convincing. I would recommend for publication. I also suggest the authors to share their algorithms on some public resources since the algorithms are important to the manuscript.

Response: Thanks very much for your comments and valuable suggestions. We have shared our algorithms on the public resources.

Reviewer: 2

Comments:

In this revised manuscript, all my concerns have been addressed, I would like to recommend it for publication in its current form.

Response: We sincerely thank the reviewer for the positive comments.

Reviewer: 3

Comments:

The comments from the reviewers have been well considered and the manuscript is now valuable to be published.

Response: Thanks for this positive comment.